# Low Temperature Ice Nucleation of Sea Spray and Secondary Marine Aerosols under Cirrus Cloud Conditions

Ryan J. Patnaude[1], Kathryn A. Moore[1], Russell J. Perkins[1], Thomas C. J. Hill[1], Paul J. DeMott[1], and Sonia M. Kreidenweis[1]

[1]Department of Atmospheric Science, Colorado State University, Fort Collins, 80521, USA

*Correspondence to:* Ryan Patnaude (ryan.patnaude@colostate.edu)

**Abstract.** Sea spray aerosols (SSA) represent one of the most abundant aerosol types on a global scale and have been observed at all altitudes including the upper troposphere. SSA has been explored in recent years as a source of ice nucleating particles (INPs) in cirrus clouds due to the ubiquity of cirrus clouds and the uncertainties in their radiative forcing. This study expands upon previous

works on low temperature ice nucleation of SSA by investigating the effects of atmospheric aging of SSA and the ice nucleating activity of newly formed secondary marine aerosols (SMA) using an oxidation flow reactor. Polydisperse aerosol distributions were generated from a Marine Aerosol Reference Tank (MART) filled with 120 L of real or artificial seawater and were dried to very low relative humidity to crystallize the salt constituents of SSA prior to their subsequent freezing, which was measured using a Continuous Flow Diffusion Chamber (CFDC). Results show that for both primary SSA (pSSA), and the aged SSA and SMA

(aSSA+SMA) at temperatures > 220 K, homogeneous conditions (92 – 97 % relative humidity with respect to water ($RH_w$)) were required to freeze 1 % of the particles. However, below 220 K, heterogeneous nucleation occurs for both pSSA and aSSA+SMA at much lower $RH_w$, where up to 1 % of the aerosol population freezes between 75–80 % $RH_w$. Similarities between freezing behaviors of the pSSA and aSSA+SMA at all temperatures suggest that the contributions of condensed organics onto the pSSA or alteration of functional groups in pSSA via atmospheric aging did not hinder the major heterogeneous ice nucleation process at

these cirrus temperatures, which have previously been shown to be dominated by the crystalline salts. Occurrence of 1 % frozen fraction of SMA, generated in the absence of primary SSA, was observed at/near water saturation below 220 K, suggesting it is not an effective INP at cirrus temperatures, similar to findings in the literature of other organic aerosols. Thus, any SMA coatings on the pSSA would only decrease the ice nucleation behavior of pSSA if the organic components were able to significantly delay water uptake of the inorganic salts, and apparently this was not the case. Results from this study demonstrate the ability of lofted

primary sea spray particles to remain an effective ice nucleator at cirrus temperatures, even after atmospheric aging has occurred over a period of days in the marine boundary layer prior to lofting. We were not able to address aging processes under upper tropospheric conditions.

## 1. Introduction

Cirrus clouds are present at temperatures below 235 K and are composed entirely of ice. With widespread global coverage, cirrus

clouds have a significant effect on the Earth's radiation budget (Baker, 1997; Chen et al., 2000; Liou, 1986; Sassen et al., 2008; Wylie et al., 1994). The sign of the radiative effect of cirrus clouds is controlled by their microphysical properties (Jensen et al., 1994; Schumann et al., 2012; Zhang et al., 1999), which are directly affected by the formation pathway. Ice nucleation in cirrus clouds at temperatures < 235 K can proceed via two basic processes, homogeneous or heterogeneous nucleation. The homogeneous (or spontaneous) nucleation pathway is the process in which condensed water containing dissolved solute particles freezes at

sufficiently high supersaturations and low enough temperatures (DeMott et al., 2003; Koop et al., 2000a), while heterogeneous

nucleation is controlled by the presence of ice nucleating particles (INPs), which lower the energy barrier required for the activation of a critical ice embryo (Hoose and Möhler, 2012; Vali et al., 2015). These two pathways will ultimately compete depending on the relative humidity (RH) and temperature conditions, which are driven by vertical motions (Kärcher et al., 2022), and the INP concentrations, which will define the lower RH threshold required for the onset of heterogeneous freezing. Homogeneous nucleation represents the upper bound RH for freezing and will depend upon the water activity of the dilute aerosol solutions and temperature (Schneider et al., 2021). Homogeneous nucleation occurs at higher ice supersaturations ($SS_i$, relative humidity with respect to ice ($RH_i$) minus 100 %) than heterogeneous nucleation. Heterogeneous nucleation may follow a number of different freezing pathways depending on the temperature, RH, and aerosol composition. The immersion freezing mechanism under cirrus cloud conditions occurs when an INP initiates freezing from within a sufficiently dilute aqueous solution droplet at relative humidity lower than that required for homogeneous freezing conditions at a given temperature. Deposition nucleation occurs at RHs between ice and water saturation, where water vapor initiates freezing directly onto an INP. However, in recent years this mechanism has been debated and some consider the freezing event to be a result of pore condensation and freezing (PCF), where water vapor condenses into pores or cracks on the surface of an INP (David et al., 2019; Marcolli, 2014).

A wide range of aerosol types may act as INPs in cirrus clouds. Global airborne observations of cirrus cloud residual particles found that mineral dust and metallic particles made up the highest fraction of cirrus ice residuals (IRs) (Cziczo et al., 2004, 2013; Cziczo & Froyd, 2014). Laboratory studies using continuous flow diffusion chambers (CFDCs) have confirmed these observations, indicating that mineral dust and metal oxides may freeze via the immersion or deposition nucleation mode between -40 ºC and -60 ºC depending on the relative humidity and particle surface area (Archuleta et al., 2005; DeMott et al., 2003; Kanji & Abbatt, 2010). While black carbon (BC) and secondary organic aerosols (SOA) are found in abundance in the upper troposphere (Froyd et al., 2010; Schill et al., 2020; Schwarz et al., 2017) and appear to dominate IRs in cirrus contrails (Petzold et al., 1998), there is uncertainty in their ability to act as effective cirrus cloud INPs. For example, Mahrt et al. (2020) showed some enhanced ice nucleating ability of BC particles in cirrus clouds after cloud processing, but freshly emitted BC particles were ineffective. Another study found that BC particles were only effective at initiating heterogenous nucleation at cirrus temperatures if they were > 100 nm and contained surface pores to freeze via PCF (Mahrt et al., 2018). While some SOA have shown the ability to freeze heterogeneously at very low active fractions (Wolf et al., 2020a), it is generally assumed that the majority of SOA freeze via homogeneous nucleation. Ammonium sulfate is abundant in the atmosphere and may also act in solid form as a cirrus INP, and although most studies have shown sulfate to freeze via heterogeneous nucleation at low active fractions (Abbatt et al., 2006; Baustian et al., 2010; Wise et al., 2010), their abundance may contribute meaningful impacts on cirrus cloud formation (Beer et al., 2022). Sea salt particles, on the other hand, have been shown to make up a relatively large fraction, up to ~25 %, of the IRs in cirrus clouds, especially when collected over the ocean (Cziczo et al., 2013). Sea spray aerosols (SSA) are abundant in the atmosphere (Vignati et al., 2010) and may be lofted from the marine boundary layer to the upper troposphere via deep convection and detrainment from cirrus anvils, where they are available for subsequent cirrus ice nucleation.

Marine aerosols are generally separated into primary SSA and secondary marine aerosols (SMA). Primary SSA are produced by wind stress on the ocean surface, which causes wave breaking and the production of bubble-bursting film and jet drops, comprised of inorganic sea-salt and organic matter (O'Dowd and De Leeuw, 2007). SMA, on the other hand, may be produced via gas-to-particle conversion of the gas phase species that are emitted from the ocean, which includes dimethyl sulfide (DMS), a biproduct of marine phytoplankton, and other volatile organic compounds (VOCs) (Fitzgerald, 1991; O'Dowd and De Leeuw, 2007). DMS and VOCs may also alter the composition of primary SSA via condensation onto existing particles. SMA have been shown to be comprised primarily of non-sea-salt (nSS) sulfates and other secondary organic species, produced primarily from the oxidation of DMS, which can dominate the total submicron aerosol mass (Fitzgerald, 1991; O'Dowd et al., 2004). However, observations in

the North Atlantic showed wide variability in the submicron organic mass depending on season (Saliba et al., 2020). SMA may play an important role in cloud formation in oceanic regions, specifically by modulating the CCN concentrations. CCN concentrations in marine environments have been found to be directly related to DMS emissions (Berresheim et al., 1993; Pandis et al., 1994). Ship based measurements in the Southern Ocean found higher concentrations of CCN containing more biogenic sulfates and organics closer to Antarctica, where there is a greater source of biogenic emissions during austral summer (Sanchez et al., 2021). More recently, laboratory studies have focused on the impact of ocean emissions on atmospheric chemistry (Mayer et al., 2020a; Prather et al., 2013) and have utilized oxidation flow reactors (OFRs) (Peng and Jimenez, 2020) to study the formation of SMA and the aging processes of the primary SSA. For example, Mayer et al. (2020b) found that SMA generated from natural seawater using an OFR plays a dominant role in the formation of cloud droplets, compared to primary SSA. Despite the attention given to studying the impact of SMA on CCN concentrations and the formation of liquid clouds, the ability of SMA to take part in ice nucleation in cirrus clouds is not well understood.

In contrast to SMA, the ice nucleating ability of primary SSA at cirrus temperatures has received more attention in recent years. A number of laboratory studies have observed heterogeneous nucleation behavior of NaCl, synthetic sea-salt, and SSA generated from real seawater at temperatures < 235 K (Schill & Tolbert, 2014; Wagner et al., 2018; Wagner & Möhler, 2013; Wise et al., 2012). Most significantly, the inorganic components of SSA have been found to freeze heterogeneously at high fractions (~10 %) below 225 K (Patnaude et al., 2021; Wagner et al., 2021), high enough to potentially compete with more conventionally effective INPs (i.e., mineral dust). In Patnaude et al. (2021) we posited that our ice nucleation results for simpler model SSA were a result of the competition between full deliquescence and immersion freezing. Those freezing results at temperatures > 220 K were characteristic of homogeneous freezing, in agreement with other previous studies (Schill and Tolbert, 2014; Wagner et al., 2018), which was likely due to the full deliquescence of the salt components around ~75 % $RH_w$, as defined by Tang and Munkelwitz, (1993) that occurred prior to the onset of immersion freezing. Below 220 K, SSA particles have been found to initiate heterogeneous nucleation at/near the range of deliquescence relative humidities (DRHs) for NaCl and SSA. Both Schill and Tolbert, (2014) and Wagner et al. (2018) concluded that at temperatures below 220 K, the phase state of their SSA particles was characterized by a solid core with a brine layer shell, and speculated that heterogeneous freezing proceeded via the immersion pathway. Patnaude et al. (2021) further discussed the reason for the transition between homogeneous and heterogeneous pathways around 220 K, which also happened to occur at the intersection of the DRH and PCF freezing parameterization from Marcolli et al. (2014) for a specific pore size. However, due to the lower DRHs for the additional salt components such as magnesium and calcium (Tang et al., 1997) the PCF pathway is less likely. The exact heterogeneous freezing mechanism of SSA and cause of the transition between freezing pathways remains unresolved.

While the non-sea-salt (nSS) components of primary SSA have been shown to act as heterogeneous INPs in the mixed-phase regime (≳ 235 K, presence of liquid and ice) (Ickes et al., 2020; McCluskey et al., 2018b), until recently less was known regarding the freezing of nSS components at cirrus temperatures. Possible types of primary marine organics may include bacteria, fatty acids, proteins, and saccharides (Wolf et al., 2019). A few recent studies have investigated the ice nucleating behavior of organic material at cirrus temperatures and found particles with a higher organic carbon fraction were more likely to freeze via heterogeneous nucleation below 235 K (Wagner et al., 2021; Wolf et al., 2019, 2020b). Additionally, Wagner et al. (2021) speculated on whether organic rich particles present in seawater would be aerosolized with a more realistic generation method and thus have the potential to act as heterogenous INPs. The inclusion of additional organic components in primary SSA may have implications for ice nucleation both below and above 220 K. Clearly, organics from seawater that are contained in primary SSA, and/or those added to the aerosol through atmospheric processing of SSA, are responsible for a major proportion of the ice nucleation activity of SSA in the mixed-phase cloud regime (McCluskey et al., 2018a), and a question has remained as to how effective these organics are as

INPs below 238 K. Organic aerosols have been shown to form a glassy state under cirrus cloud conditions (Berkemeier et al., 2014; Ignatius et al., 2016; Knopf et al., 2018; Koop et al., 2011; Murray et al., 2010) and depending on the glass transition temperature and DRH, these particles may have heterogeneous ice nucleating ability at temperatures > 220 K, where the salt components of SSA would have already dissolved. It is unknown how organic rich particles would compete with the salt components of SSA below 220 K, but based on the results in Kasparoglu et al. (2022), the contribution of glassy SOA may be relatively minor. These studies illustrate the potential of the organic material and inorganic salts in seawater to initiate heterogeneous nucleation at cirrus temperatures. However, the pathway of SSA from the marine boundary layer to the upper troposphere where cirrus clouds reside would likely modify both the inorganic and organic components of these aerosols due to atmospheric oxidation. The question of whether atmospheric oxidation alters the ice nucleating ability of SSA at cirrus temperatures remains unanswered.

In this study, expanding upon the work in Patnaude et al. (2021), the low temperature ice nucleating ability of SSA, generated using a Marine Aerosol Reference Tank (MART) to resemble realistic ocean–atmosphere emissions, is investigated. Using an Aerodyne potential aerosol mass OFR, we examined the impact of atmospheric oxidation on freezing behavior of primary SSA at cirrus temperatures. In addition, the low temperature freezing behavior of SMA produced from gas-phase emission of seawater was investigated. Finally, we compared primary SSA and SMA generated from real seawater and an artificial seawater product to infer the roles of the organic material and inorganic sea salts on heterogeneous ice nucleation at cirrus temperatures.

**2 Methods**

A general overview of the experimental setup is shown in Figure 1. The following sections provide descriptions of each component of the aerosol generation and processing instrumentation and the design of the experiments.

**2.1 Seawater collection and preparation**

For this study, a Marine Aerosol Reference Tank (MART), originally developed by the NSF Center for Aerosol Impacts on Chemistry of the Environment (CAICE, Prather et al., (2013)), was used for generation of an atmospherically relevant size distribution of SSA by mimicking wave breaking in the ocean (Mayer et al., 2020a; Stokes et al., 2013). The MART was filled with 120 L of 1) natural seawater (SW) collected at the Scripps pier in La Jolla, CA, 2) Neomarine (Brightwell Aquatics), a commercially available artificial seawater (ASW) product (3.5 % by weight) or 3) deionized water (DI). The ASW was mixed outside of the MART in 5-gallon plastic (LDPE/LLDPE) cubitainers, filtered through a TOC+HEPA filter and into the MART, then recirculated through the filter using a peristaltic pump for ~24 hours to remove residual organic carbon. The seawater was shipped overnight at room temperature to Colorado State University in the same cubitainers, which had been acid washed prior to use in mixing the ASW, then thoroughly rinsed with deionized water before SW collection. During seawater collection, the water was filtered through a 50-μm food-grade woven polyester mesh, which was pre-cleaned with methanol and 5 % hydrogen peroxide and rinsed with deionized water. Unlike the ASW, the SW was not filtered before being added to the MART in order to retain biological material. Since the MART was located indoors, it was equipped with a VIPARSPECTRA full spectrum aquarium grow light to provide broad spectrum light to the microorganisms inside the tank during experimental days and turned off at night. The grow light was set to a realistic PAR (photosynthetically active radiation) quantity of ~175 μmol m$^{-2}$ to keep the microorganisms active and not stimulate a bloom. A hollow aluminum coil connected to a water-cooling bath was inserted into the MART to cool the seawater to ~15 ºC to mimic spring ocean temperatures in San Diego. However, during experiments the water warmed (< 5 ºC) due to the heat exhaust from the CFDC chillers located near the MART and cooled again overnight. The centrifugal pump used to recirculate and plunge the water in the MART can damage microorganisms due to strong pump shear (Mayer et al., 2020a) and

this effect can be highly selective and may change the composition of the biologic community. It is unclear how this affects the microorganisms that are directly emitted as SSA and subsequent ice nucleating ability, but this impact has existed in prior MART
studies of freezing by nascent SSA (McCluskey et al., 2018a). MART plunging was turned on shortly after the icing stage of the CFDC (discussed in section 2.3, about 30 minutes before measurement) to ensure aerosol concentrations had equilibrated in all the lines prior to the ice nucleation experiments.

## 2.2 Aerosol generation and characteristics

Aerosol generation from the MART is accomplished by intermittently plunging a sheet of water on top of the water surface,
producing a plume of bubbles with a distribution similar to those measured in the ocean. The subsequent polydisperse aerosol size distribution produced from the bursting of bubbles at the water surface closely resembles that which is produced from ocean wave breaking in the ambient environment (Mayer, et al., 2020a). The sealed headspace of the MART containing both aerosols and any emitted gases was modestly pressurized by a zero-air generator (ZAG) pumping 7 liters per minute by volume (LPM$_v$) into the MART and forcing out the sample air.
The aerosol sample line split in two after exiting the MART with one sample line sent through two silica gel diffusion dryers to the upstream set of sizing instruments to confirm the nascent SSA number size distributions produced by the MART. The other sample line was passed through the OFR, whether powered to produce oxidation or not (see below), for further processing and characterization of ice nucleation behavior, to ensure aerosol losses were consistent during all the experiments. The OFR generates high concentrations of O$_3$ and OH radicals to oxidize particles using two UV lamps at wavelengths $\lambda$ = 254 nm and $\lambda$ =185 nm
(Mayer et al., 2020b). The UV lamp type used in this study was GPH436T5L/VH/4P 90/10, which emits radiation of the 185 nm wavelength at 0.6 % of the intensity of the 254 nm wavelength (Rowe et al., 2020). The OH exposure in the OFR was calibrated using the change in carbon monoxide (CO) concentrations inside the OFR, using the same temperature and RH conditions as during the experiments. The change in CO versus light intensity of the lamps uses the CO + OH rate coefficient ($k_{OH+CO}$ ~1.48 * 10$^{-13}$ cm$^3$ molecules$^{-1}$ s$^{-1}$). The residence time in the OFR was ~2.4 min and the average OH exposure in the experiments was ~6.31 * 10$^{11}$
molecules sec/cm$^3$ which translated to ~4–6 days of aging under typical atmospheric conditions (OH = 1.5 *10$^6$ molecules cm$^{-3}$). In addition, the RH inside the OFR was maintained > 60 % to ensure OH chemistry dominated the oxidation processes and minimize O-singlet reactions. Because the air was humidified inside the OFR, the oxidation reactions that occurred would mimic those which occur in the marine boundary layer.

Sample air exiting the OFR first passed through a Nafion dryer, which used ~10 LPM$_v$ of counter flow in its dry air circuit to
dehydrate the sample air. After drying, sample air was passed through a diffusion dryer packed with Carulite, an O$_3$ destroying catalyst, to remove residual O$_3$ and then passed through a final 4-way split: 1) molecular sieves dryer and CFDC, 2) second set of sizing instruments, 3) ice spectrometer filter (IS), and 4) RH monitor and excess flow vent (Figure 1). The IS was used to measure INPs in the mixed-phase temperature regime (-38 – 0 ºC) (Hiranuma et al., 2015; Vali, 1971). The RH monitor confirmed the aerosol stream RH was generally <10 % for all experiments, so aerosol size distribution measurements and IS filter collections
were considered to be dry. Prior to particles entering the CFDC they passed through a coil cold trap immersed in the inner wall chiller, which further chilled and dried particles and incoming sample air to a frost point at least 5 K below the CFDC measurement temperature, corresponding to an ambient temperature RH of ~0 % (see Kasparoglu et al. (2022)).

Three different aerosol types were generated and investigated for both natural and artificial seawater, depending on whether the MART plunging and OFR were active (see Table 1). The generation, oxidation, sizing and flow configuration was the exact same
for both water types. The three aerosol types were 1) primary or freshly emitted SSA (pSSA), 2) "aged" (or oxidized) SSA and SMA (aSSA+SMA, henceforth), and 3) SMA (SMA-only). We hypothesize SMA is formed mostly through the oxidation of DMS,

similar to Mayer et al., (2020b), who showed that sulfates contributed to > 50 % of the mass of SMA generated from real seawater in a MART. Two duplicate CFDC experiments were conducted for both pSSA and aSSA+SMA measurements, while only one experiment was conducted for SMA-only. Between the artificial and natural seawater experiments, the MART was cleaned and filled with DI water and a "blank" experiment was conducted to test background contamination. This experiment was conducted the same as the SMA experiments where the MART plunging was turned off and only the MART headspace was sampled.

Size distributions of the aerosol samples were measured using a combination of a Scanning Mobility Particle Sizer (SMPS, TSI models 3080, 3081, 3010; 12-600 nm) and an Aerodynamic Particle Sizer (APS, TSI model 3321; 0.52 – 20 μm). For this study, two sets of aerosol sizing instruments were used, one located upstream of the OFR, and one located downstream of the OFR, Nafion, and Carulite dryers (see Figure 1). Measurements from the downstream set were used to quantify the sample losses that occurred inside the OFR and dryers. Size distributions comparing both upstream and downstream sets of sizing instruments can be found in the Supplementary Figure S1 and represent an average of the SMPS and APS scans over the course of each ice nucleation experiment, ~4 hours. The APS measurements were converted from aerodynamic to geometric diameter using a dynamic shape factor and particle density of SSA of 1.05 and 1.9 g cm$^{-3}$, respectively, consistent with values from observations and those applied in previous studies (Brock et al., 2019; Patnaude et al., 2021; Zieger et al., 2017).

The phase state of the particles before entering the CFDC remains uncertain and could affect the heterogeneous ice nucleating ability of the aerosol population. Several previous studies have analyzed the temperature- and humidity-dependent phase states of NaCl particles, since they are often used as a proxy for SSA. Previous studies have found that the phase state of NaCl after efflorescence depends on the temperature at which efflorescence occurred, with anhydrous NaCl observed for efflorescence temperatures above 273 K (Bartels-Rausch et al., 2021; Koop et al., 2000b; Peckhaus et al., 2016; Wagner et al., 2012). Since the majority of aerosol drying, including the efflorescence of the salt components of pSSA and aSSA+SMA, occurred at room temperature (Figure 2), it is believed the salts would be in the anhydrous form. However, the phase state and morphology of co-emitted organic particles or coatings on the pSSA and aSSA+SMA are less well understood at low temperatures, and the exact structure was uncertain in this study. This is especially difficult to know for the aSSA+SMA particles, due to the exposure to oxidation in the OFR. It is assumed that particles generated from SW contain high fractions of insoluble organic particles or salt particles mixed with organic carbon below 500 nm, similar to previous lab-generated SSA (Bertram et al., 2018; Kaluarachchi et al., 2022a; Prather et al., 2013). In addition, DeMott et al. (2023) showed particle morphologies of laboratory-generated SSA that include organic coatings in the submicron size range, both before and after similar use of an OFR for oxidation studies. The MART was demonstrated to produce substantially similar size distributions and SSA compositions compared to a more natural wave breaking process for bubble bursting (Prather et al., 2013; Stokes et al., 2013), so no bias in organic content in comparison to natural SSA production is expected. During cooling, the RH$_w$ would be low enough such that organic particles or coatings may form a glassy state (Ignatius et al., 2016; Knopf et al., 2018; Koop et al., 2011), as shown by the shaded green region in Figure 2, which represents the glass transition conditions for sucrose (Zobrist et al., 2008).

Aerosols that contain organic material may either enhance heterogeneous nucleation (Wilson et al., 2012) or have no effect (Kasparoglu et al., 2022). For example, organic coatings on mineral dust particles have been shown to suppress heterogeneous nucleation toward higher RH at cirrus temperatures (Möhler et al., 2008), depending also on the coating thickness or the fractional coverage of the particles, while another study found organic coatings on mineral dust had no effect on immersion freezing between 233–253 K regardless of the coating thickness (Kanji et al., 2019). At warmer temperatures > 233 K, organic coatings may be in a less viscous or liquid-like phase state, while at colder temperature may become semi-solid or glassy (Charnawskas et al., 2017), which may explain the differences in ice nucleation behavior of particles with organic coatings in those studies. Specifically, the

inclusion of glassy coatings on the pSSA or aSSA+SMA particles may inhibit water uptake and delay dissolution of the salt components until higher $RH_w$ is reached.

**2.3 Ice nucleation detection using the CFDC**

Detection of low temperature ice nucleation for all experiments was carried out using a CSU CFDC, which had been modified from the design used in previous studies (Archuleta et al., 2005; DeMott et al., 2015; DeMott et al., 2009) to reach temperatures down to 190 K. A full description of the modifications made to the CFDC to reach lower temperatures, and a complete derivation of calculations and measurement uncertainties can be found in Patnaude et al. (2021). The CFDC is made up of two vertically-oriented concentric cylinders consisting of a "cold" inner wall and "warm" outer wall. The chamber temperature and relative humidity (RH) are controlled by holding the inner and outer walls at different temperatures, producing temperature and water partial pressure gradients. This approach can produce a very broad range of RH conditions as low as 0 % with respect to ice and up to double digit supersaturations with respect to water (Rogers, 1988). The walls are covered with a thin layer of ice by chilling the column to 246 K and filling, then draining, the column with deionized water prior to ice nucleation experiments. The flow rates during experiments were 4 $LPM_v$ sheath flow and 1 $LPM_v$ sample flow (calculated in the CFDC interior). The low sample flow translates to a longer residence time inside the CFDC, allowing for more ice crystal growth and thus less ambiguous optical detection of ice crystals versus much smaller unfrozen aerosol particles. An optical particle counter (OPC) is located immediately downstream of the CFDC column at its base and freezing of aerosol particles is distinguished by analyzing the number of particles, presumed to be ice crystals, above a certain size.

In these experiments, the CFDC was operated to perform $RH_w$ "scans" wherein the column temperature was held constant and the $RH_w$ gradually increased until a specified frozen fraction was achieved, in this case 10 %. During an individual scan the inner wall temperature ($T_{IW}$) was held constant while the outer wall temperature ($T_{OW}$) was gradually raised, increasing the column supersaturation. Once 10 % frozen fraction was reached, the $T_{IW}$ was raised 5 K and the $T_{OW}$ was lowered until the frozen fraction dropped below 0.1 %, and the cycle was repeated. The thresholds of ice fraction for the $RH_w$ scans required 10 continuous seconds above 10 % or below 0.1 % in order to reset the scans. Note these activated fractions were calculated based on particle counts in channels above a selected OPC channel to define ice counts during the experiments, as opposed to the more accurate method for calculation of ice fraction that accounted for the background aerosol distribution (described below). Using 10 % and 0.1 % as endpoints of the $RH_w$ scans resulted in a broad range in RHs being covered. By the time 10 % of the particles had nucleated, the $RH_w$ was at above homogeneous freezing conditions (Koop et al., 2000a) and it was not necessary to continue raising the $RH_w$. The lower threshold of 0.1 % for 10 continuous seconds was sufficient to reset the scans as the $RH_w$ was then well below the deliquescence $RH_w$ for NaCl and SSA (~74 %; Tang and Munkelwitz, (1993)), with the exception of $RH_w$ at warmer temperatures > 225 K, where the RH scans did not reach below ~75 % but we did not expect heterogeneous freezing conditions.

Careful attention was given to the challenging identification of slow-growing ice crystals, without incorrectly classifying large aerosols as ice, when using a polydisperse aerosol size distribution and without the presence of an aerosol impactor upstream of the CFDC to remove large particles from the sample stream. In this study, it was not sufficient to only use a single OPC cut size at all temperatures for identifying ice crystals for two reasons: 1) the larger salt particles may undergo significant hygroscopic growth into the size range of ice crystals at RHs above their deliquescence point , and 2) Patnaude et al. (2021) demonstrated the lack of size dependence on the freezing of SSA, meaning the smaller salt particles would likely freeze along with the larger particles and may not be counted if they are below the OPC cut size. Instead, we analyzed the OPC size distributions for increasing RHs, as shown in Figure 3, and identified the background aerosol distributions when the RH is lower, but after deliquescence has likely occurred. Figure 3a shows the OPC size distributions at -45 °C for pSSA generated from SW and shows little change in the

distribution until the RH reaches 99 % (green line) where a significant number of larger particles emerged, signifying ice nucleation was occurring. At the lower RHs (< 95 %) the salt particles in the seawater would have already deliquesced and taken up water, and therefore these distributions represent the background aerosol population that would need to be removed to count the total number of ice crystals. The lack of change in the size distributions between 80–90 % may be due to smaller SSA growth factors below 95 % (Tang et al., 1997), or kinetic limitations at these temperatures. Figure 3b shows another example of the OPC distributions for measurements at -68 °C. In this case, there was a much more considerable shift in the distributions with increasing RH, more likely due to ice nucleation and to a lesser extent from hygroscopic growth. Figure S3 shows a timeline of the particle counts from lower OPC channels, similar to Figure 3 from Kong et al. (2018) but does not show the sharp step where deliquescence occurred. This may be due to several reasons, such as the lower vapor pressures at a colder temperature (213 K versus 233 K in Kong et al. (2018)), or that in this study we used a polydisperse aerosol distribution so smaller particles taking up water would shift into the same OPC channels that larger particles should be shifting out of toward larger sizes. However, both the pSSA and aSSA+SMA did show some decrease in the number of particles as the $RH_w$ increased above the deliquescence RH (~74 %) indicating more gradual water uptake and slower deliquescence transition at lower temperatures.

When comparing the two temperatures (Figures 3a and b), the distributions in Figure 3b below 75 % (blue and red lines) closely match the lower RH distributions in Figure 3a, indicating they are likely the background aerosols at this temperature and that the additional shift to larger sizes for RHs > 75 % was a result of particles freezing. Based on this analysis we used bin channel 40 on the OPC as the lower (more conservative) threshold for ice, which from previous calibrations represents particles ~2 μm for a flow rate of 5 $LPM_v$, where roughly 0.1 % of the dry aerosol particles are present (see Figure S1). Therefore, the total number of ice crystals is calculated as the difference between the OPC spectra and background aerosol distributions summed above channel 40 (denoted by the blue shaded region in Figure 3). This method considers both the nucleation of smaller particles and eliminates larger particles that have not nucleated ice. The inferred mechanisms of freezing at these temperatures will be discussed below.

## 3. Results and Discussion

### 3.1 Particle size distributions and alteration via OFR oxidation

Data from the SMPS and APS downstream of the OFR were merged into a complete size distribution, as shown in Figure 4 for all experiments. The solid lines indicate aerosols generated from real seawater and the dashed lines are those generated from artificial seawater. For sizes < 200 nm the aSSA+SMA distributions were dominated by secondary particle formation, as indicated by their similarity to the SMA-only experiment. This is consistent with previous work that showed only a small fraction of the submicron aSSA+SMA number distribution generated from a MART originated from the pSSA (Mayer et al., 2020b; Prather et al., 2013). Mayer et al., (2020b) also suggested that new particle nucleation was favored over condensation in the OFR due to the high OH concentrations and fast oxidation rates. The SW size distributions had higher concentrations of aSSA+SMA than the ASW at sizes < 100 nm, which may indicate additional gas phase emissions from the seawater, capable of oxidizing to condensable species. In general, both aSSA+SMA distributions agree with the pSSA distributions at larger sizes, however this occurs at different aerosol diameters for the ASW and SW, where the distributions begin to converge at ~200 nm and 1 μm for the ASW and SW, respectively. In addition, the higher concentration of aSSA+SMA compared to pSSA from natural SW between 200 nm and 1 μm (Figure 4b) suggests some modification of the pSSA. This could occur through a number of different factors, including gas-phase condensation, changes to the seawater microbial activity altering emissions, or minor changes to particle generation due to surface tension or temperature.

Although there were higher number concentrations of aSSA+SMA between 200 nm and 1 µm, there was not a discernable shift towards larger sizes, which would be expected after condensation from the gas phase onto the pSSA. In addition, the low concentrations of pSSA would limit condensation of organics in the OFR and would favor new particle formation. The estimated condensation sink timescales were calculated and shown in the supplemental material, indicating that the condensation sink timescale for the nucleation mode aerosols (SMA-only) was ~3 minutes compared to ~11 minutes for the pSSA. Therefore, it is likely the nucleation mode particles scavenged the majority of the condensable material in the OFR and minimized condensation onto the pSSA. However, prior OFR studies for similar pSSA loadings indicated modest increases in organic volume fractions in pSSA as detected by atomic force microscopy (AFM) following similar oxidation exposures using the same OFR (Kaluarachchi et al., 2022a). These changes were accompanied by modifications in particle phase state in the water subsaturated regime, as well as hygroscopicity, for submicron pSSA. DeMott et al. (2023) discussed how similar OFR studies on laboratory generated pSSA led to apparent changes in organic functionalization, at least as determined by AFM for submicron particles. Raman spectroscopy did not detect functional changes following oxidation in that study, but this was inferred to be a consequence of the emphasis on the 1 µm and larger particle regime for the Raman spectroscopic studies, for which organic volume fractions are already quite small and signal to noise becomes an issue. Finally, DeMott et al. (2023) noted that OFR processing of pSSA led to a decrease in INP concentrations by a factor of a few times in the temperature regime >243 K, where heterogeneous ice nucleation has been shown to be initiated by the organic components in very small fractions of pSSA. It has been unknown if this alteration of ice nucleation at higher temperatures translates to impacts at cirrus temperatures. Further, it is also not known whether changes induced by oxidization of the organics present in the pSSA, or the changes in ice nucleating activity due to organics added via condensation or functional alteration might affect the heterogeneous nucleation process that was inferred to be stimulated by crystalline salts at temperatures below 220 K (Patnaude et al., 2021).

Both pSSA size distributions showed peaks around 100 nm, in agreement with other studies that generated SSA in laboratory settings (Collins et al., 2014; Patnaude et al., 2021; Quinn et al., 2015). However, there was a slight secondary mode at ~800 nm that only occurred during the real seawater experiments (Figure 4b). These two aerosol modes at 100 nm and 800 nm likely represent particles generated from film and jet drops, respectively, consistent with a previous laboratory study that generated aerosol particles from real seawater (Hill et al., 2023). Prather et al. (2013) showed that the fraction of biological particles increased for particles > 1 µm, and can be up to ~20 % of particles > 2 µm. Therefore, this apparent mode of particles ~800 nm may represent additional aerosolized biologic material such as bacteria, gels and viruses, that may be enriched with organic particles (Hill et al., 2023). These organic-rich biological particles would not exist in the ASW, and may explain lower concentrations in the aSSA+SMA generated from ASW compared to SW between 200 nm and 1 µm. The differences in the ~800 nm mode could also be due to modifications in droplet generation mechanisms, through small differences in temperature, RH, and surfactant content (Stokes et al., 2013).

For the blank (background) tests, in which the MART was filled with only DI water and the OFR was turned on but MART plunging turned off (black line), it was found that there was also secondary particle formation, likely a result of VOCs emitted from the acrylic material of the MART walls. It is likely that the outgassing of acrylic tank materials and subsequent aerosol formation from those compounds in the OFR occurred in all oxidation experiments. The outgassing may have varied with water temperature, which increased throughout each experimental day, and could contribute to some of the differences observed in the aSSA+SMA and SMA-only size distributions. For example, during the pSSA and aSSA+SMA experiments, particle concentrations below 100 nm were higher later in the day due to the warming of the water with time (Figure S2). During the blank test the water was not pre-cooled as in the SW and ASW experiments due to time constraints, which likely led to higher VOC emissions and may explain the high secondary particle concentrations observed during the blank measurements. Hence, the proportion of

secondary particles or condensable material formed in the SMA-only and aSSA+SMA cases that was contributed from direct gas-phase emission from the SW and ASW, compared to other organics from the acrylic outgassing, cannot be determined. Future studies will need to probe SMA ice nucleation by using non-plastic materials. We will nevertheless assume the role of SMA can be interpreted from the ice nucleation experiments, as will be discussed in the subsequent results sections.

### 3.2 Analysis of ice nucleation results

Figure 5a shows a time series of CFDC scans for a pSSA experiment. The first three scans, which occurred below 220 K showed a gradual increase in ice particle counts (light blue markers) with increasing $RH_w$, starting around 65 %. The final three scans beginning ~218 K and above, showed a more modest increase in ice particle counts until the $RH_w$ reached nearly 100 %, resulting in a very sharp increase in ice particle counts. The first three scans indicate particles freezing via heterogeneous nucleation due to the slow and gradual increase of ice particles, with the initial formation of ice particles below 75 % $RH_w$. In contrast, the last three scans showed very few ice particles below 85 % $RH_w$, but then had a sudden and dramatic spike in ice particles when the $RH_w$ was > 90 %. The fourth scan may be at/near the transition between the two freezing mechanisms, as there were some increased ice particles counts at $RH_w$ < 80 %, but we also observed the sharp spike in ice particles when the $RH_w$ was near water saturation. These results were consistent with previously reported freezing behaviors of NaCl and sea spray particles at temperatures between 205 K and 235 K (Patnaude et al., 2021; Wagner et al., 2018) who showed time series of freezing indicative of heterogeneous freezing below ~218 K and homogeneous freezing above 218 K. Ice nucleation experiments illustrating CFDC scans for SMA-only generated from sampling the headspace over real seawater are shown in Figure 5b. In this case, all the scans showed little ice formation (< 1 $L^{-1}$) until the $RH_w$ reached > 100 %, where there was a sudden and rapid onset of freezing of nearly all the aerosol particles. These cases illustrate that SMA froze through a homogeneous freezing mechanism, as there was almost no ice formed until the $RH_w$ exceeded conditions for expected homogeneous freezing (Koop et al., 2000a). The CFDC $RH_w$ scanning procedure was modified for the SMA cases and is the reason for the slower increase in $RH_w$ after ice is initially formed. Additionally, most of the SMA particles formed are below the minimum size of detection of the OPC (~300 nm), therefore the sharp increase in total particles (black line) is mostly a result of the particles freezing and growing to sizes that can be detected by the OPC.

The ice nucleating ability of pSSA and aged aSSA+SMA generated from the MART is examined in Figure 6. The fractions of frozen particles at 1 % and 5 % are shown in T – $RH_w$ space. The frozen fraction was calculated by dividing the CFDC OPC-measured ice concentrations (discussed previously) by the integrated total particle counts measured in the second set of sizing instruments (post-OFR losses). The sizing instruments provided a total aerosol count, since the OPC may not have detected the smallest particles at its lower size bound (~0.3 microns) even at RH exceeding expected deliquescence RH, particularly for the SMA experiments. Note when using this method, 10 % ice fraction was no longer observed in any of the experiments and is the reason for presentation of 5 % instead. For calculations of frozen fraction for the aSSA+SMA experiments, the pSSA size distributions were used with the assumption that pSSA would freeze first. The colored markers denote 1 % (blue) and 5 % (green) frozen fraction, where the filled and open markers denote the SSA generated from natural and artificial seawater, respectively. The reference lines indicate ice saturation (black dashed), homogeneous freezing calculated from Koop et al., (2000a) (black solid), and the estimated pore condensation RH for an 11 nm pore size (Marcolli, 2014) (yellow solid), as in Patnaude et al. (2021). For temperatures > 218 K, the majority of both pSSA and aSSA+SMA particles froze at conditions near the threshold expected for homogeneous freezing (Koop et al., 2000a). Interestingly, both 1 % and 5 % frozen fractions of pSSA, and 5 % frozen fraction of aSSA+SMA, closely follow the upper bound of the expected homogeneous freezing parameterizations for aqueous sulfuric acid particles (Schneider et al., 2021) as temperatures decrease from 230 K to 217K. 1 % frozen fractions of aSSA+SMA particles followed the Koop et al. (2000a) average homogeneous freezing conditions more closely, within the range of uncertainties for

calculation of $RH_w$ in the CFDC (4 %) (Richardson, 2009). It should be noted that this is not a random uncertainty, that if there is error in the calculated $RH_w$ it will be the same for all measured points. Results for 0.1 % frozen fraction (Figure S4) indicated that a small fraction of the aerosol population still froze via heterogeneous nucleation at temperatures warmer than 218 K, with $RH_w$ ~5-10 % lower than the 1 % observations. However, they still fell within the lower uncertainty bounds suggested by Schneider et

al. (2021) for freezing of aqueous sulfuric acid. The 0.1 % frozen fraction for particles generated from natural seawater also occurred at lower $RH_w$ (by up to 5 %) than for those from artificial seawater, perhaps indicating the inclusion of a more active subset of organic INPs in the real seawater. One might expect these more effective immersion freezing INPs to be present in concentrations that could be inferred from higher temperature freezing data, and if contained within the overall SSA population, their freezing conditions should parallel homogeneous freezing conditions but with a water activity offset (Archuleta et al., 2005).

During the pSSA experiments the total aerosol concentrations were ~150 cm$^{-3}$, therefore 0.1 % of the aerosol population would be 150 L$^{-1}$. When extending the immersion freezing line to water saturation (water activity = 1), as shown in Figure S5, the number of INPs per liter from the IS spectra at those temperatures, ~0.005 L$^{-1}$ at 251 and ~0.5 L$^{-1}$ at 245 K for SW and ASW, respectively, was much lower than reflected by 0.1 % freezing of all particles at lower temperatures. Therefore, we did not observe the expected number of heterogeneous freezing INPs based on the ice spectrometer spectra (not shown). We do not understand the exact

mechanism for the freezing of the lower fractions of particles in the temperature regime where homogeneous freezing dominates. In other words, the 0.1 % frozen fraction is not consistent (higher than) the expected contribution of immersion freezing of the more specialized INPs that are present in SSA for mixed-phase temperatures (when using the water activity freezing concept). Below 218 K, the onset of freezing for both pSSA and aSSA+SMA occurred at much lower $RH_w$, ~70 %, for both the artificial and real seawater experiments. This shift from homogeneous freezing to heterogeneous freezing for the major proportion of particles

at temperatures below ~220 K was consistent with several previous experiments investigating the freezing of NaCl and SSA particles (Ladino et al., 2016; Patnaude et al., 2021; Schill & Tolbert, 2014; Wagner et al., 2018; Wolf et al., 2019; 2020a). Additionally, for both water types, the onset of heterogeneous freezing occurred near the estimated deliquescence RHs of NaCl and the full deliquescence of SSA (blue shaded region), similar to the results in previous studies (Patnaude et al., 2021; Wagner et al., 2018, 2021). The deliquescence RH of NaCl was estimated for colder temperatures from Tang & Munkelwitz, (1993), and was

lowered ~4 % RH for SSA following Wagner et al. (2018) with the assumption that the additional salts in seawater would deliquesce at a lower RH. The observations in Figure 6 also demonstrated that the alteration of the pSSA via oxidation (either through condensed organics or modification of the functional groups) did not modify the heterogeneous ice nucleation of pSSA. Both the pSSA and aSSA+SMA results showed that a transition from primarily homogeneous freezing to heterogeneous nucleation occurred at ~218 K.

Similar to Patnaude et al. (2021), the pathway for heterogeneous nucleation and the transition between homogeneous and heterogeneous below 220 K remains unresolved. Deposition freezing remains an unlikely freezing pathway for pSSA and aSSA+SMA (at 1 % frozen fraction) at/near the DRH as water uptake of minor salt constituents would begin at lower RHs (Schill and Tolbert, 2014; Tang et al., 1997; Wagner et al., 2018). The competition between full deliquescence and immersion freezing or PCF to explain this transition in freezing pathways was also discussed in Patnaude et al. (2021) but bears repeating. While the 1%

frozen fraction of both pSSA and aSSA+SMA occurred at near the PCF line, we do not believe this to be the likely freezing pathway for two reasons: 1) the PCF line shown in Figure 6 was approximated for an 11 nm pore size, and there is little evidence this is an realistic size for SSA and 2) the SSA particles would need to be fully dry to retain surface pores, which was not likely the case at 75 % $RH_w$ where 1 % frozen fraction occurred. At temperatures above ~220 K, particle deliquescence would occur at a lower $RH_w$ than the expected value for immersion freezing (see Patnaude et al., (2021) Figure 8), thus favoring full dissolution and

homogeneous nucleation at much higher RHs. At lower temperatures (< 220 K), the particles may have developed a brine layer,

which has been observed in two previous studies (Schill and Tolbert, 2014; Wagner et al., 2018), and ice nucleation may have proceeded via the immersion pathway. Therefore, the most likely heterogeneous freezing pathway for pSSA and aSSA+SMA was immersion freezing.

In contrast to the ice nucleation results for pSSA and aSSA+SMA, the SMA generated by sampling and oxidizing the gases in the headspace of the MART, over both artificial and natural seawater samples, showed no indication of heterogeneous nucleation at any temperature (Figure 7; 1 % frozen fraction shown). For SMA generated from a combination of emissions and acrylic VOCs from both water types, 1 % frozen fraction did not occur until $RH_w$ had almost reached water saturation, well above the expected freezing threshold for homogeneous nucleation. Due to the rapid onset of ice particle formation as the $RH_w$ was increased beyond the homogeneous freezing threshold, 1 % was the lowest fraction that could be reliably detected by the OPC. These results support the conclusion that pure SMA does not freeze via heterogeneous nucleation at cirrus temperatures, similar to previous studies that analyzed the ice nucleating ability of non-marine SOA in the cirrus regime (Kasparoglu et al., 2022).

## 4. Conclusions

In this study, a MART provided a representative method for generating polydisperse SSA for comparison to our prior studies of monodisperse NaCl and SSA particle freezing at cirrus temperatures (Patnaude et al., 2021). Despite the added complexity from generating a polydisperse SSA distribution, the impact on heterogeneous ice nucleation behavior was minimal. This was illustrated by the striking similarities in ice nucleation results between the pSSA from this study and the results from Patnaude et al. (2021), which both determined heterogeneous ice formation with activated fractions above 1 % only occurred at temperatures of 218 K or below. The results from this current study also corroborate the association of the transition from homogeneous to heterogeneous nucleation at these cirrus temperatures with the presence of crystallized (undissolved) salts in pSSA, and with no demonstrated size dependence. Based on the similarities between freezing results in this study and Patnaude et al. (2021) and Wagner et al. (2018), the dominant heterogeneous freezing mechanism below 218 K was likely the immersion freezing pathway.

The results in this study specifically address questions raised by Wagner et al. (2021): will a higher number of ice-active particles be generated from a sea spray chamber where highly organic particles may be formed (Wilson et al., 2015; Wolf et al., 2020b); or, if highly organic-rich particles are present, would these particles be aerosolized when generated in a more realistic manner? It is possible that organic particles in the pSSA from real seawater may contribute a small additional subset of the INP population and freeze more efficiently at warmer temperatures, as shown by the observed 0.1 % frozen fraction at much lower $RH_w$ above 220 K than the 1 % or 5 % frozen fractions (Figure S5). Additionally, the 0.1 % frozen fraction from natural SW has $RH_w$ up to 5 % lower than the 0.1 % results from ASW, which may also be a result of organic enrichment. Organic-rich particles may not deliquesce until higher RHs (>80 %), allowing for heterogeneous nucleation under these conditions, whereas pure salt particles would be fully dissolved and would not freeze until the homogeneous freezing threshold is reached. However, even if these are organic-enriched or internally mixed particles and represent a subset of the INPs in marine aerosols at cirrus levels, they are vastly outnumbered by the inorganic salt particles that appear to dominate heterogeneous ice nucleation at these temperatures.

No prior studies have investigated how atmospheric aging may impact SSA ice nucleation at cirrus temperatures. This study showed that the freezing of pSSA particles after undergoing atmospheric aging and the formation of an additional SMA population via new particle formation (simulated by use of an OFR) was not changed compared to the freshly emitted pSSA particles, as demonstrated by the similar results obtained with both pSSA and aSSA+SMA. There are a few possible reasons for the similar freezing behaviors between these two particle types: 1) the newly formed particles scavenged the majority of condensable material produced in the OFR and only modest amounts of secondary organics condensed to the pSSA, 2) the air stream remained

humidified when entering the OFR and the pSSA particles were thus likely in an aqueous state. Whether oxidation of their organic content could proceed in the wetted or partially wetted state, and whether changes to the pSSA organic content would be similar to that observed when processed in a dry crystalline state is unknown, or 3) the addition of SMA coatings to the pSSA and/or alteration of organic components of the pSSA did not alter the crystallization behaviors, nor did they hinder the water uptake by the inorganic salts. Nevertheless, the inclusion of organic material on the pSSA would likely only hinder the heterogeneous nucleation ability that remains dominated by the crystalline salts. Future studies using more realistic or at least slower atmospheric aging processes, such as what may occur in a typical smog chamber, may better represent the possible organic coatings on pSSA. Additionally, oxidation of pSSA at cold temperatures ($< 233$ K) and after efflorescence may produce different freezing results than this study due to oxidation under different phase states and morphologies. These results also indicate that secondary particle formation from the gas-phase emissions of natural seawater does not generate effective INPs at cirrus temperatures, as the SMA-only case did not reach 1 % frozen fraction until the $RH_w$ had almost reached water saturation. Thus, SMA may only be considered a relevant source for ice nucleation in cirrus clouds if they are exposed to a pathway that begins via activation as a CCN into cloud droplets at lower altitudes, lofted to cirrus temperatures in deep convection, and then freeze homogeneously (liquid-origin cirrus, see Krämer et al., (2020) and Luebke et al., (2016)).

This study could not distinguish whether the experiments were affected by any of the following: changes to the phase state or morphology of the pSSA particles following oxidation; whether fragmentation of aerosol-phase organic compounds occurred; or if glassy states were induced. Exploring these possibilities would require that associated compositional studies be aligned with similar experiments (Mayer et al., 2020b; Prather et al., 2013; DeMott et al., 2023; Kaluarachchi et al. 2022a, 2022b), but these results suggested that the strongest heterogeneous freezing behaviors noted at temperatures below 218 K were dominated by the salt components even after atmospheric aging. One remaining caveat to this study regarding the SMA and aSSA+SMA results is the uncertainty of the contribution from the seawater versus outgassing from the MART to the mass of secondary material produced, which should be addressed by future studies that might also include compositional and phase state data.

We have shown SSA can nucleate ice via heterogeneous nucleation below 220 K. Airborne observations have found concentrations of sea salt between $10^{-4} - 10^{-1}$ µg m$^{-3}$ in the upper troposphere (Bian et al., 2019; Murphy et al., 2019), and based on the results shown herein a large proportion of those salt particles should retain their ice nucleating ability even after the atmospheric aging that would occur during their ascent to cirrus levels. Cirrus clouds are ubiquitous in the upper troposphere and represent a large source of uncertainty in determining Earth's radiative budget. In addition, cirrus clouds that form via heterogeneous nucleation have also been shown to represent a net positive radiative forcing (Krämer et al., 2020), thus demonstrating a potential indirect radiative effect of SSA on our climate system.

**Author contributions**

R. Patnaude, P. DeMott, S. Kreidenweis contributed to the development of the ideas, quality controlled the data, and wrote the majority of the manuscript. R. Patnaude contributed to the analysis of the data. K. Moore and R. Perkins performed the majority of the experimental setup and contributed to the experimental design and execution. T. Hill contributed to the cleaning, preparation, and collection of the ice spectrometer filters. All co-authors contributed to reviewing and editing of the manuscript.

**Competing interests**

The authors declare that they have no conflict of interest.

## Code and data availability

The data from this study are available at: doi:10.5061/dryad.kprr4xhb2

## Acknowledgements

This work was supported by the National Science Foundation (NSF) through the NSF Center for Aerosol Impacts on Chemistry of the Environment (NSF-CAICE), Award CHE-1801971. We would like to acknowledge Matt Pendergraft and Raymond Leibensperger III for their assistance in the seawater collection and shipment to Colorado. We would like to acknowledge Kathryn J. Mayer for assistance with the setup and calibration of the OFR. K. Moore acknowledges support by an NSF Graduate Research Fellowship under Grant 006784. Any opinions, findings, and conclusions or recommendations expressed in this material are those of the author(s) and do not necessarily reflect the views of the National Science Foundation.

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

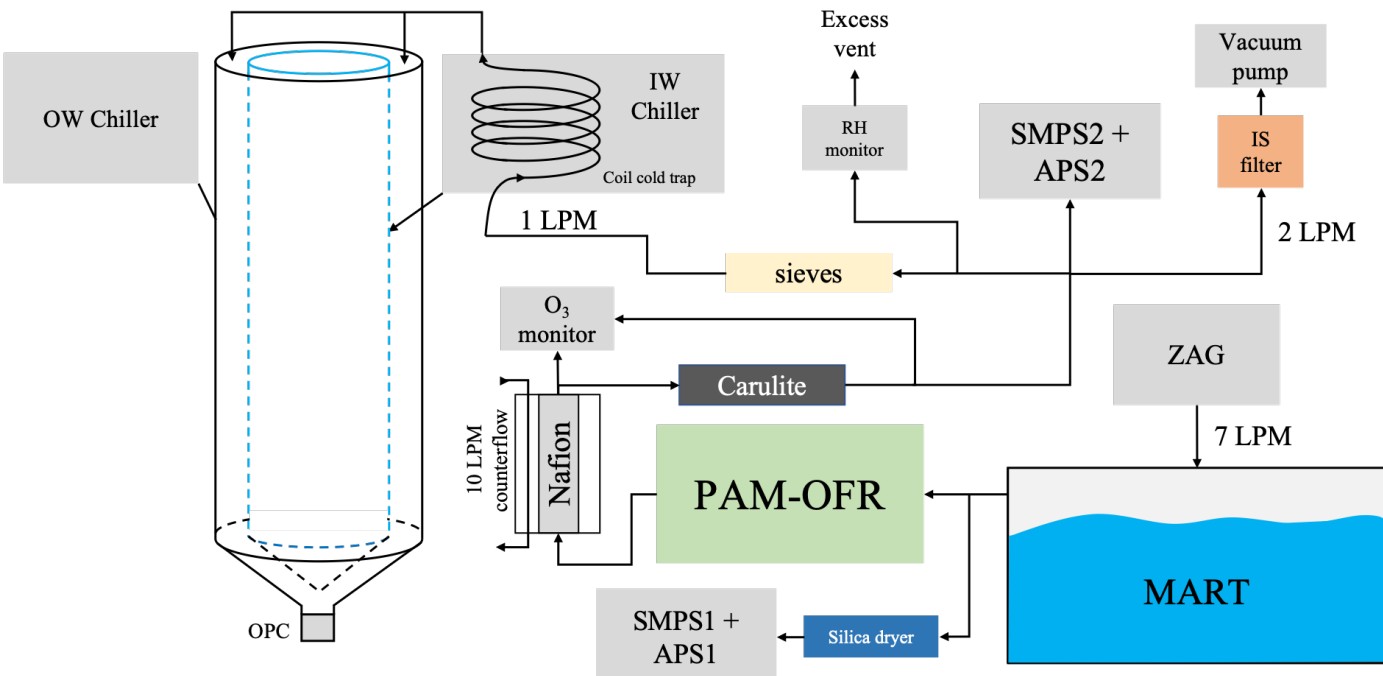


**Figure 1.** Schematic of the experimental setup for the MART studies. Flow configuration remained the same for all aerosol generation methods and water types.

**Table 1.** Descriptions of experimental aerosol generated from the MART for each water type.

| Generated aerosol types (Water type) | OFR lamp operation | MART plunging |
|---|---|---|
| pSSA (ASW, SW) | OFF | ON |
| aSSA+SMA (ASW, SW) | ON | ON |
| SMA only (ASW, SW, DI) | ON | OFF |


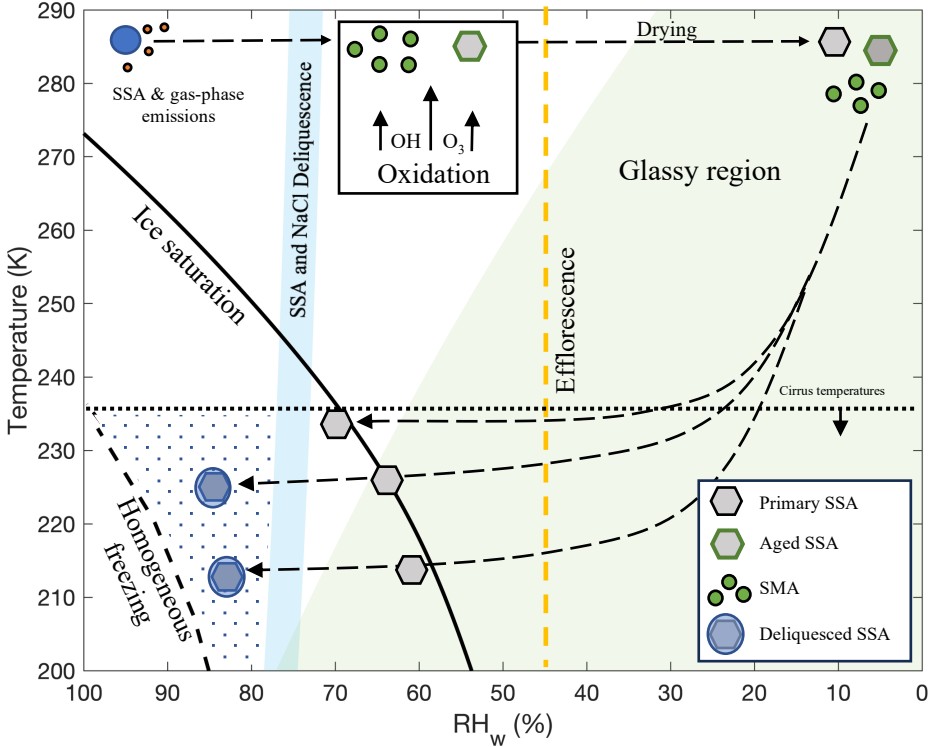

**Figure 2.** Expected trajectory and phase state of the pSSA, aSSA+SMA and SMA particles for CFDC experiments, modified from Patnaude et al. (2021). Orange dashed line is the expected efflorescence line for NaCl on the basis of the parametrization of anhydrous NaCl and extrapolated to cirrus temperatures (Tang & Munkelwitz, 1993). The blue shaded region represents the range of possible deliquescence RHs for NaCl and SSA, using the parameterization from Tang & Munkelwitz, (1993) for NaCl extrapolated to colder temperatures as the upper bound and shifting it down 4 % RH$_w$ for SSA similar to Wagner et al. (2018). The long dashed black lines follow the path of aerosol particles as through drying, cooling, and CFDC scans at different temperatures. The blue circles represent aqueous solutions, gray hexagons represent effloresced pSSA aerosols, and the light blue circles with embedded hexagons represent fully deliquesced particles. The gray hexagons with green outlines and green circles represent the aSSA+SMA and SMA particles, respectively. Lines indicating ice saturation and predicted homogeneous freezing conditions are also denoted. The dotted region represents conditions where aerosol particles experience ice supersaturated conditions and relative humidities that exceed their deliquescence point. The green shaded region represents conditions below the glass transition curve of sucrose from Zobrist et al. (2008).

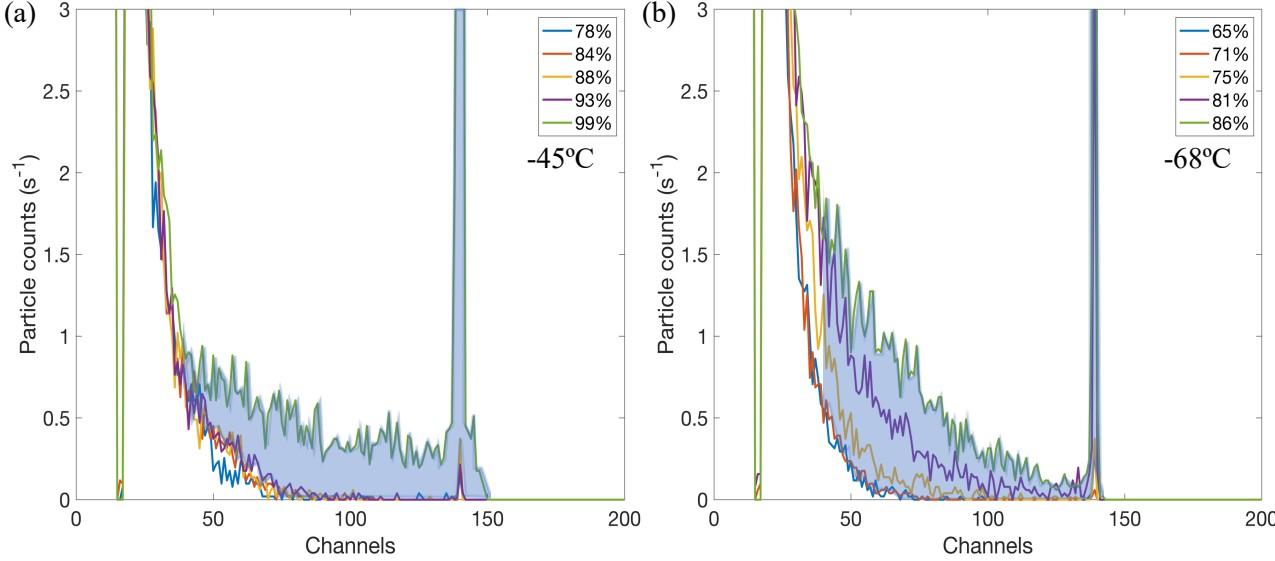

**Figure 3.** CFDC OPC size distributions of pSSA for increasing RH at (a) -45 ºC and (b) -68 ºC during the first of two real seawater experiments. The blue shaded region represents the difference between OPC counts at higher RHs and counts of background aerosol distributions at lower RHs above channel 40, denoting what would be distinguished as ice particles.

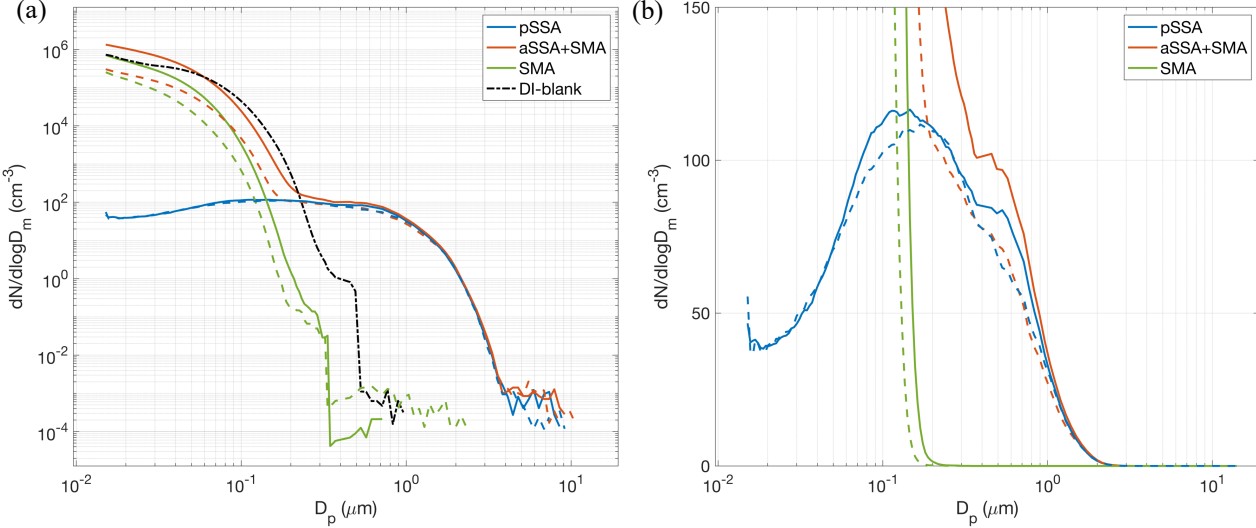

**Figure 4.** (a) Particle size distributions combined from the SMPS and APS measured downstream of the OFR and (b) the same as (a) but zoomed in a linear axis. The solid lines denote aerosols generated from SW and the dashed lines from ASW. The DI-blank experiment is denoted by the black dashed-dotted line. Size distributions represent a time-average of the SMPS and APS scans measured throughout the entire ice nucleation experiment.

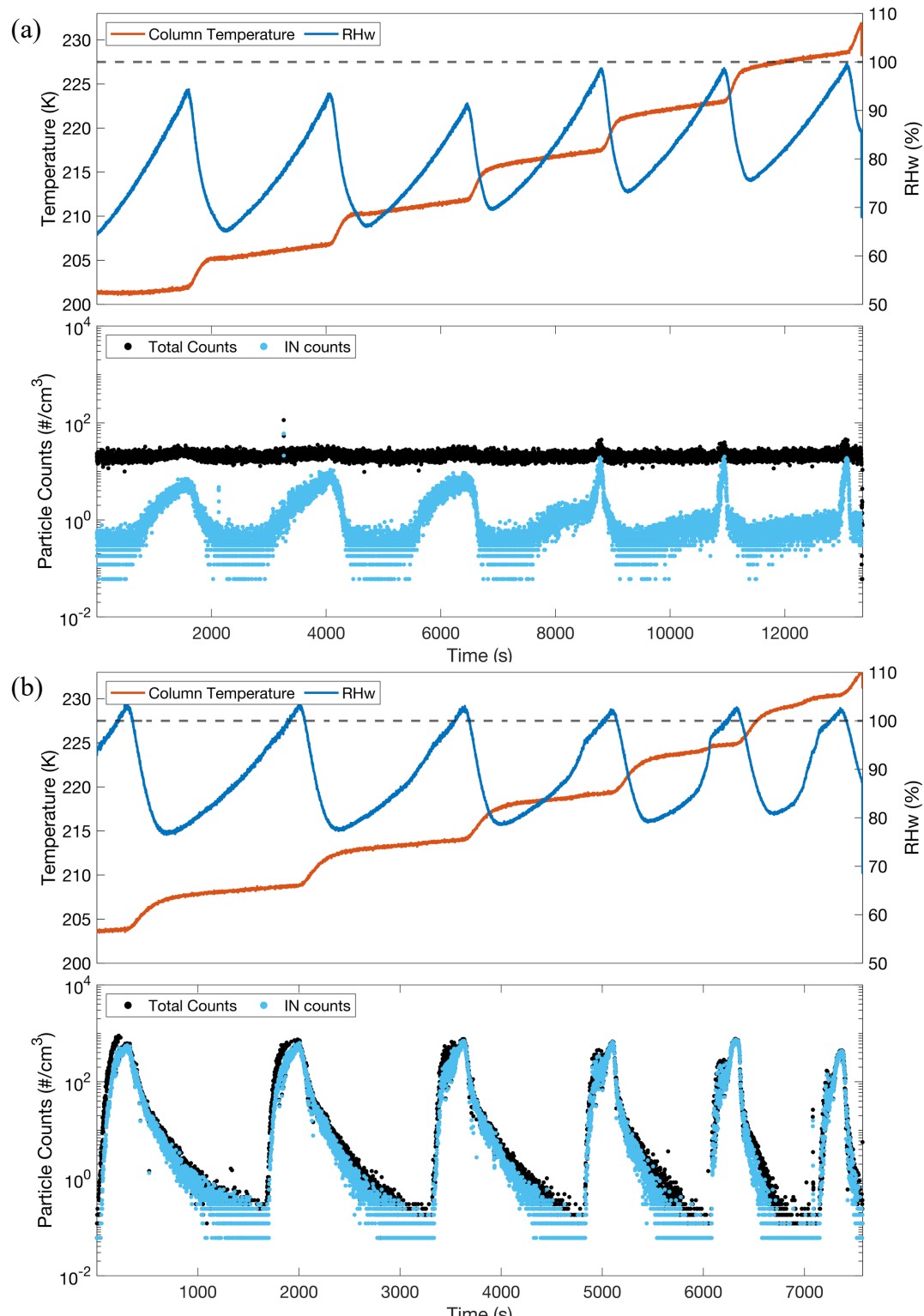

**Figure 5.** Time series of CFDC scans for (a) pSSA and (b) SMA experiments generated from real seawater beginning at 204 K. The CFDC column temperature is represented by the red lines, and the blue lines represent the RH$_w$. The black and light blue markers indicate the numbers of total particles counted in the OPC and those that are considered ice crystals, respectively. The black dotted horizontal lines in the top panels of (a) and (b) represent water saturation.

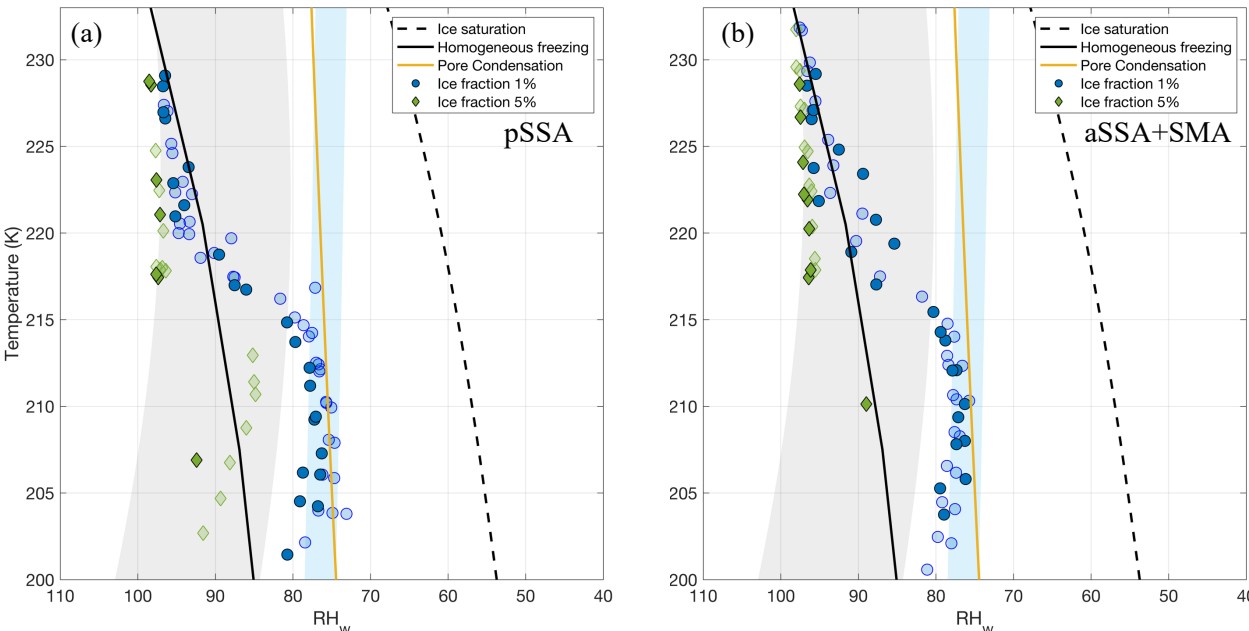

**Figure 6.** Fraction of frozen particles for (a) pSSA and (b) aSSA+SMA particles. The filled markers denote particles generated from the real seawater and the open markers from the artificial seawater. The blue and green markers indicate 1 %, and 5 % frozen fractions, respectively. The reference lines indicate ice saturation (black dashed), the parameterized $RH_w$ for pore condensation in an 11 nm pore (Marcolli, 2014) (solid yellow), and the expected threshold for the onset of homogeneous freezing (Koop et al., 2000a) (solid black). The blue shaded region represents the range of possible deliquescence RHs for NaCl and SSA, using the parameterization from Tang & Munkelwitz, (1993) for NaCl extrapolated to colder temperatures as the upper bound and shifting it down 4 % RH for SSA similar to Wagner et al., (2018). The gray shaded region represents the range of uncertainty for the homogeneous freezing parameterization for aqueous sulfuric acid particles as presented in Schneider et al. (2021b).

830

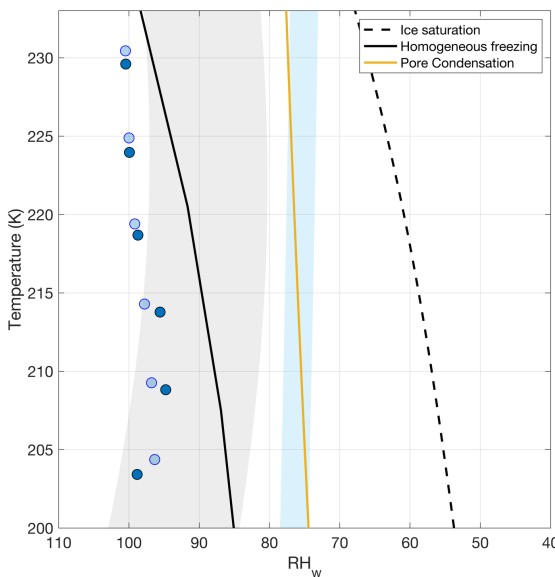

**Figure 7.** Similar to Figure 6, but only showing 1 % frozen fraction of SMA particles, where the filled and open markers denote SMA generated by sampling the headspace over real and artificial seawater, respectively.