# Peer review of "Low Temperature Ice Nucleation of Sea Spray and Secondary Marine Aerosols under Cirrus Cloud Conditions"

_EGUsphere, 2023_

## Referee Comment (RC1)

**General comments:**

In this paper, the authors present their findings on the ice-nucleating particle (INP) characteristics of the Sea Spray Aerosol (SSA), which were generated from a Marine Aerosol Reference Tank (MART). Additionally, the researchers explored the impact of atmospheric aging on these characteristics. Interestingly, they found no observable effect of the atmospheric aging. On the whole, the study is methodologically sound. However, I must express three major concerns as well as a few specific issues related to the content, which I will delve into more deeply below. Despite these concerns, the study remains intriguing and offers valuable contributions to the broader scientific community's understanding of the INP characteristics of SSA. There is no doubt that with necessary revisions, the work will be worthy of publication. Nonetheless, it is imperative to note that major revisions are required to elevate the study to its full potential.

**Three major issues:**
1. The freezing mechanism at temperature < 220K
The elucidation of the nucleation mechanism in SSA remains a significant and yet unresolved scientific query. The authors of the current study, intriguingly, appear to circumvent direct discussion of the low-temperature nucleation dynamics of SSAs. They opt instead to vaguely encapsulate the complex phenomena using the generic term 'heterogeneous freezing.' The data put forth in this paper, particularly as illustrated in Figure 5, presents a compelling view. It appears to document a transition from homogeneous to heterogeneous nucleation as temperatures descend towards 220 degrees. Nucleation observed under these chillier conditions within a range spanning from water to ice saturation. This behavior should ideally be defined as deposition nucleation, however, Figure 5 sheds light on the temperature interval wherein the pore condensation freezing (PCF) manifests itself. A striking alignment is observed between the nucleation occurring below 215 degrees and the PCF. This concurrence seemingly substantiates the notion that SSA nucleation under colder conditions could indeed be characterized by the PCF.

2. The phase state of the SSA
The main objective was to compare the INP characteristics of the pure and aged SSA, however, the reviewer was concerned the state of particles could influence the results. In this study, the measurement of humidity was performed before the coil cold trap, maintaining a controlled relative humidity at 10% under ambient temperature conditions. However, this level of water vapor pressure can escalate from a few thousand to tens of thousands supersaturation with respect to ice at 220K. Consequently, it is imperative for the authors to consider the dwell time within the coil cold trap and the Continuous Flow Diffusion Chamber (CFDC). Furthermore, it would be beneficial to generate estimations of the phase state prior to its entry into the CFDC.

3. The setup of the experiment
Currently, the sample air directly enters the oxidation flow reactor after exiting from the MART instrument. It is suggested that the sample air should be dehumidified before

passing through the oxidation flow reactor. This is because, during liquid-phase oxidation, the crystallization of SSA solution droplets into crystals may not affect its surface structure. However, if oxidation occurs in the solid phase, the pores on the surface of the SSA particles might be filled, thereby affecting its ice nucleation properties.

**Specific comments:**

P1, Line 16: I suggest use the supersaturation with respect to ice instead of RH to evaluate the onset of the ice nucleating forms.

P3, Sec 2.1: The artificial seawater was filter through a TOC+HEPA filter to remove the insoluble particles, what about the natural seawater? There might be some dust and biological particles inside the natural seawater which have influence on the INP measurement.

P5, Line 151-152: I was wondering why there were so many particles during the "blank experiment" with DI water. Does that mean there were contamination of the MART and sampling tubes? Thus, I strongly suggest do "blank experiment" before and after each experiment.

P5, Line 155: TSI models 3080, remove the 3081 and 3010, which is the model of DMA and CPC.

P5, Line 160: This paragraph is confusing and very hard to understanding.

Line 195-295, when discussing the influence of organics, some studies (Ignatius, et al., 2016; Knopf et al., 2018; Tian et al., 2022) found organic aerosol (likely secondary) could be glassy (Koop et al., 2011) and efficient heterogeneous ice nuclei under the condition of low RH, which could be referenced to support the point that organics itself may serve as INP.

P7, Sec2.4: The IS show the mixed phase regime (-38–0 ℃) INP concentration which inconsistent with the main theme of this study, and the results was shown in the supplement. The author need to carefully consider whether to retain this section of content.

**References:**

Koop, T., Bookhold, J., Shiraiwa, M., Pöschl, U. Glass transition and phase state of organic compounds: dependency on molecular properties and implications for secondary organic aerosols in the atmosphere. *Phys. Chem. Chem. Phys.,* 2011, 13, 19238-19255.

Knopf, Daniel A., Alpert, Peter A., Wang, B. B. The role of organic aerosol in atmospheric ice nucleationL: A review. *ACS Earth Space Chem*, 2018, 2, 168-202. DOI: 10.1021/acsearthspecechem.7b00120.

Tian, P., Liu, D. T. Bi, K. et al. Evidence for anthropogenic organic aerosols contributing to ice nucleation. *Geophysical Research Letters,* 49, e2022GL099990.

https://doi.org/10.1029/2022GL099990.

Ignatius, K., Kristensen, T. B., Jarvinen, E., at al. Heterogeneous ice nucleation of viscous secondary organic aerosol produced from ozonolysis of α-pinene. *Atmos. Chem. Phys.,* 2016, 16, 6495-6509.

---

## Author Comment (AC1)

**ACP Response to RC1 for *Low Temperature Ice Nucleation of Sea Spray and Secondary Marine Aerosols under Cirrus Cloud Conditions***
Format: The reviewers' comments are quoted in italics
Line number in the response refers to the revised manuscript with tracked changes.
Quotation in red color stands for revised/added text in the revised manuscript
Responses in blue

**General comments**
*In this paper, the authors present their findings on the ice-nucleating particle (INP) characteristics of the Sea Spray Aerosol (SSA), which were generated from a Marine Aerosol Reference Tank (MART). Additionally, the researchers explored the impact of atmospheric aging on these characteristics. Interestingly, they found no observable effect of the atmospheric aging. On the whole, the study is methodologically sound. However, I must express three major concerns as well as a few specific issues related to the content, which I will delve into more deeply below. Despite these concerns, the study remains intriguing and offers valuable contributions to the broader scientific community's understanding of the INP characteristics of SSA. There is no doubt that with necessary revisions, the work will be worthy of publication. Nonetheless, it is imperative to note that major revisions are required to elevate the study to its full potential.*

We would like to start by thanking the reviewer for their thoughtful comments and contributions to this manuscript. We will address both the major and minor below.

**Three major issues:**
*1.The freezing mechanism at temperature < 220K*
*The elucidation of the nucleation mechanism in SSA remains a significant and yet unresolved scientific query. The authors of the current study, intriguingly, appear to circumvent direct discussion of the low-temperature nucleation dynamics of SSAs. They opt instead to vaguely encapsulate the complex phenomena using the generic term 'heterogeneous freezing.' The data put forth in this paper, particularly as illustrated in Figure 5, presents a compelling view. It appears to document a transition from homogeneous to heterogeneous nucleation as temperatures descend towards 220 degrees. Nucleation observed under these chillier conditions within a range spanning from water to ice saturation. This behavior should ideally be defined as deposition nucleation, however, Figure 5 sheds light on the temperature interval wherein the pore condensation freezing (PCF) manifests itself. A striking alignment is observed between the nucleation occurring below 215 degrees and the PCF. This concurrence seemingly substantiates the notion that SSA nucleation under colder conditions could indeed be characterized by the PCF.*

The pathways of heterogeneous nucleation were not resolved in this study or Patnaude et al., (2021). However, based on the recommendations of multiple reviewers we have added additional background on the dominant freezing pathways discussed in previous studies in the Introduction, and have inferred the immersion freezing pathway as the most likely heterogeneous freezing mechanism in this study as discussed in the Results section 3.2. and Conclusions.

L117-L129: "In Patnaude et al. (2021) we posited that our ice nucleation results for simpler model SSA were a result of the competition between full deliquescence and immersion freezing. Those freezing results at temperatures > 220 K were characteristic of homogeneous freezing, in agreement with other previous studies (Schill and Tolbert, 2014; Wagner et al., 2018), which was likely due to the full deliquescence of the salt components around ~75 % $RH_w$, as defined by Tang and Munkelwitz, (1993) that occurred prior to the onset of immersion freezing. Below 220 K, SSA particles have been found to initiate heterogeneous nucleation at/near the range of deliquescence relative humidities (DRHs) for NaCl and SSA. Both Schill and Tolbert, (2014) and Wagner et al. (2018) concluded that at temperatures below

220 K, the phase state of their SSA particles was characterized by a solid core with a brine layer shell, and speculated that heterogeneous freezing proceeded via the immersion pathway. Patnaude et al. (2021) further discussed the reason for the transition between homogeneous and heterogeneous pathways around 220 K, which also happened to occur at the intersection of the DRH and PCF freezing parameterization from Marcolli et al. (2014) for a specific pore size. However, due to the lower DRHs for the additional salt components such as magnesium and calcium (Tang et al., 1997) the PCF pathway is less likely. The exact heterogeneous freezing mechanism of SSA and cause of the transition between freezing pathways remains unresolved."

L720-L734: "Similar to Patnaude et al. (2021), the pathway for heterogeneous nucleation and the transition between homogeneous and heterogeneous below 220 K remains unresolved. Deposition freezing remains an unlikely freezing pathway for pSSA and aSSA+SMA (at 1 % frozen fraction) at/near the DRH as water uptake of minor salt constituents would begin at lower RHs (Schill and Tolbert, 2014; Tang et al., 1997; Wagner et al., 2018). The competition between full deliquescence and immersion freezing or PCF to explain this transition in freezing pathways was also discussed in Patnaude et al. (2021) but bears repeating. While the 1% frozen fraction of both pSSA and aSSA+SMA occurred at near the PCF line, we do not believe this to be the likely freezing pathway for two reasons: 1) the PCF line shown in Figure 6 was approximated for an 11 nm pore size, and there is little evidence this is an realistic size for SSA and 2) the SSA particles would need to be fully dry to retain surface pores, which was not likely the case at 75 % $RH_w$ where 1 % frozen fraction occurred. At temperatures above ~220 K, particle deliquescence would occur at a lower $RH_w$ than the expected value for immersion freezing (see Patnaude et al., (2021) Figure 8), thus favoring full dissolution and homogeneous nucleation at much higher RHs. At lower temperatures (< 220 K), the particles may have developed a brine layer, which has been observed in two previous studies (Schill and Tolbert, 2014; Wagner et al., 2018), and ice nucleation may have proceeded via the immersion pathway. Therefore, the most likely heterogeneous freezing pathway for pSSA and aSSA+SMA was immersion freezing."

L766-L767: "Based on the similarities between freezing results in this study and Patnaude et al. (2021) and Wagner et al. (2018), the dominant heterogeneous freezing mechanism below 218 K was likely the immersion freezing pathway."

*2.The phase state of the SSA*
*The main objective was to compare the INP characteristics of the pure and aged SSA, however, the reviewer was concerned the state of particles could influence the results. In this study, the measurement of humidity was performed before the coil cold trap, maintaining a controlled relative humidity at 10% under ambient temperature conditions. However, this level of water vapor pressure can escalate from a few thousand to tens of thousands supersaturation with respect to ice at 220K. Consequently, it is imperative for the authors to consider the dwell time within the coil cold trap and the Continuous Flow Diffusion Chamber (CFDC). Furthermore, it would be beneficial to generate estimations of the phase state prior to its entry into the CFDC.*

The particles were further dried by a molecular sieves drier after the RH monitor described in the previous sentence. Additionally, they passed through a coil cold trap that was immersed in the inner wall cold trap. This chilled and dried the particles to nearly 0 % before entering the CFDC column. We have added a sentence to make this clearer.

See our response to your comment on phase state of organics below.

L211-L213: "Prior to particles entering the CFDC they passed through a coil cold trap immersed in the inner wall chiller, which further chilled and dried particles and incoming sample air to a frost point at

least 5 K below the CFDC measurement temperature, corresponding to an ambient temperature RH of ~0 % (see Kasparoglu et al. (2022))."

*3. The setup of the experiment*
*Currently, the sample air directly enters the oxidation flow reactor after exiting from the MART instrument. It is suggested that the sample air should be dehumidified before passing through the oxidation flow reactor. This is because, during liquid-phase oxidation, the crystallization of SSA solution droplets into crystals may not affect its surface structure. However, if oxidation occurs in the solid phase, the pores on the surface of the SSA particles might be filled, thereby affecting its ice nucleation properties.*

As mentioned in the text, it was intentional to keep the air somewhat humidified in order to control the oxidation chemistry in the OFR, and to maintain consistency with previous SSA experiments (Prather et al., 2013; Mayer et al., 2020b; DeMott et al., 2023). We have added an additional sentence to clarify the oxidation reactions that occurred would mimic those that occur in the boundary layer of the atmosphere, and that this may be relevant for impacts on ice nucleation. Further studies would be needed to explore oxidative impacts occurring for upper tropospheric (very dry) conditions. We may also note that we have added substantial discussion to the paper regarding expected alteration of organics based on similar prior studies that included compositional measurements, in response to comments from other reviewers.

L204-L205: "Because the air was humidified inside the OFR, the oxidation reactions that occurred would mimic those which occur in the marine boundary layer."

L784-L800: "2) the air stream remained humidified when entering the OFR and the pSSA particles were thus likely in an aqueous state. Whether oxidation of their organic content could proceed in the wetted or partially wetted state, and whether changes to the pSSA organic content would be similar to that observed when processed in a dry crystalline state is unknown, or 3) the addition of SMA coatings to the pSSA and/or alteration of organic components of the pSSA did not alter the crystallization behaviors, nor did they hinder the water uptake by the inorganic salts."

**Specific comments:**

*P1, Line 16: I suggest use the supersaturation with respect to ice instead of RH to evaluate the onset of the ice nucleating forms.*

This statement in line 16 is in reference to Figure 6, which uses $RH_w$. Because homogeneous nucleation is driven by the condensation of liquid water, we believe that $RH_w$ gives greater insight into nucleation processes than ice supersaturation (where homogeneous freezing onsets are more variable as a function of temperature).

*P3, Sec 2.1: The artificial seawater was filter through a TOC+HEPA filter to remove the insoluble particles, what about the natural seawater? There might be some dust and biological particles inside the natural seawater which have influence on the INP measurement.*

We did not filter the natural seawater because we wished to retain any biological particles that were present in the water. This was standard protocol for all previous ice nucleation studies in the literature (references already included), when using real seawater. We have added an additional statement that the seawater was not filtered.

L171-L172: "Unlike the ASW, the SW was not filtered before being added to the MART in order to retain biological material."

*P5, Line 151-152: I was wondering why there were so many particles during the "blank experiment" with DI water. Does that mean there were contamination of the MART and sampling tubes? Thus, I strongly suggest do "blank experiment" before and after each experiment.*

The blank experiments were conducted with the DI water at room temperature rather than being chilled, and exposing any gaseous species present to oxidation in the OFR. The blank experiment took place during the one day we had between the experiments with the ASW and SW, as the SW arrived and needed to be added into the MART immediately after delivery. It is discussed in the text that the reason for the high particle concentrations during this test was likely due to higher VOC emissions from the MART (acrylic) at warmer temperatures than occurred in the tests with chilled water. We leak checked the entire system after adding the DI water to the MART, and measured no particle emissions with the OFR off, therefore we do not believe it was contamination in the sampling tubes.

It would not have been possible to perform a blank test before and after each experiment due to the limited time we had with borrowed equipment because it took significant time to drain/fill the MART, which also had to be cleaned before and after changing out the water.

*P5, Line 155: TSI models 3080, remove the 3081 and 3010, which is the model of DMA and CPC.*

The SMPS system consists of a classifier, DMA, and CPC, which are the three model numbers given. The 3080 by itself is just the classifier, not a functional SMPS. There likely is a TSI model number that corresponds to the collection of these three pieces, but it is no longer easily searchable and hence not useful to use in the manuscript.

*P5, Line 160: This paragraph is confusing and very hard to understanding.*

See response to RC2 regarding this paragraph. We hope those changes have clarified this section sufficiently.

*Line 195-295, when discussing the influence of organics, some studies (Ignatius, et al., 2016; Knopf et al., 2018; Tian et al., 2022) found organic aerosol (likely secondary) could be glassy (Koop et al., 2011) and efficient heterogeneous ice nuclei under the condition of low RH, which could be referenced to support the point that organics itself may serve as INP.*

Thank you. We have added some discussion on the potential of a glassy phase state of the organic particles in both the Introduction and section 2.2:

L137-L146: "The inclusion of additional organic components in primary SSA may have implications for ice nucleation both below and above 220 K. Clearly, organics from seawater that are contained in primary SSA, and/or those added to the aerosol through atmospheric processing of SSA, are responsible for a major proportion of the ice nucleation activity of SSA in the mixed-phase cloud regime (McCluskey et al., 2018a), and a question has remained as to how effective these organics are as INPs below 238 K. Organic aerosols have been shown to form a glassy state under cirrus cloud conditions (Berkemeier et al., 2014; Ignatius et al., 2016; Knopf et al., 2018; Koop et al., 2011; Murray et al., 2010) and depending on the glass transition temperature and DRH, these particles may have heterogeneous ice nucleating ability at temperatures > 220 K, where the salt components of SSA would have already dissolved. It is unknown how organic rich particles would compete with the salt components of SSA below 220 K, but based on the results in Kasparoglu et al. (2022), the contribution of glassy SOA may be relatively minor."

L249-L258: "It is assumed that particles generated from SW contain high fractions of insoluble organic particles or salt particles mixed with organic carbon below 500 nm, similar to previous lab-generated SSA (Prather et al. 2013; Bertram et al. 2018; Kaluarachchi et al. 2022a). In addition, DeMott et al. (2023) showed particle morphologies of laboratory-generated SSA that include organic coatings in the submicron size range, both before and after similar use of an OFR for oxidation studies. The MART was demonstrated to produce substantially similar size distributions and SSA compositions compared to a more natural wave breaking process for bubble bursting (Prather et al., 2013; Stokes et al., 2013), so no bias in organic content in comparison to natural SSA production is expected. During cooling, the $RH_w$ would be low enough such that organic particles or coatings may form a glassy state (Ignatius et al., 2016; Knopf et al., 2018; Koop et al., 2011), as shown by the shaded green region in Figure 2, which represents the glass transition conditions for sucrose (Zobrist et al., 2008)."

[Figure]

**Figure 2.** Expected trajectory and phase state of the pSSA, aSSA+SMA and SMA particles for CFDC experiments, modified from Patnaude et al. (2021). Orange dashed line is the expected efflorescence line for NaCl on the basis of the parametrization of anhydrous NaCl and extrapolated to cirrus temperatures (Tang & Munkelwitz, 1993). The blue shaded region represents the range of possible deliquescence RHs for NaCl and SSA, using the parameterization from Tang & Munkelwitz, (1993) for NaCl extrapolated to colder temperatures as the upper bound and shifting it down 4 % $RH_w$ for SSA similar to Wagner et al. (2018). The long dashed black lines follow the path of aerosol particles through drying, cooling, and CFDC scans at different temperatures. The blue circles represent aqueous solutions, gray hexagons represent effloresced pSSA aerosols, and the light blue circles with embedded hexagons represent fully deliquesced particles. The gray hexagons with green outlines and green circles represent the aSSA+SMA and SMA particles, respectively. Lines indicating ice saturation and predicted homogeneous freezing conditions are also denoted. The dotted region represents conditions where aerosol particles experience ice supersaturated conditions and relative humidities that exceed their deliquescence point. The green shaded region represents conditions below the glass transition curve of sucrose from Zobrist et al. (2008).

*P7, Sec2.4: The IS show the mixed phase regime (-38–0 ℃) INP concentration which inconsistent with the main theme of this study, and the results was shown in the supplement. The author need to carefully consider whether to retain this section of content.*

We see the reviewer's point, that these data are outside the temperature regime being focused on. We have decided to remove this section from the manuscript as we also removed the supplemental figure S3 that included the ice spectrometer INP spectra (see response to RC2).

**References:**

*Koop, T., Bookhold, J., Shiraiwa, M., Pöschl, U. Glass transition and phase state of organic compounds: dependency on molecular properties and implications for secondary organic aerosols in the atmosphere. Phys. Chem. Chem. Phys., 2011, 13, 19238-19255.*

*Knopf, Daniel A., Alpert, Peter A., Wang, B. B. The role of organic aerosol in atmospheric ice nucleation: A review. ACS Earth Space Chem, 2018, 2, 168-202. DOI: 10.1021/acsearthspecechem.7b00120.*

*Tian, P., Liu, D. T. Bi, K. et al. Evidence for anthropogenic organic aerosols contributing to ice nucleation. Geophysical Research Letters, 49, e2022GL099990. https://doi.org/10.1029/2022GL099990.*

*Ignatius, K., Kristensen, T. B., Jarvinen, E., at al. Heterogeneous ice nucleation of viscous secondary organic aerosol produced from ozonolysis of α-pinene. Atmos. Chem. Phys., 2016, 16, 6495-6509.*

**New References:**

DeMott, P. J., Hill, T. C. J., Moore, K. A., Perkins, R. J., Mael, L. E., Busse, H. L., Lee, H., Kaluarachchi, C. P., Mayer, K. J., Sauer, J. S., Mitts, B. A., Tivanski, A. V, Grassian, V. H., Cappa, C. D., Bertram, T. H. and Prather, K. A.: Atmospheric oxidation impact on sea spray produced ice nucleating particles, Environ. Sci. Atmos., doi:10.1039/d3ea00060e, 2023.

Kaluarachchi, C. P., Or, V. W., Lan, Y., Madawala, C. K., Hasenecz, E. S., Crocker, D. R., Morris, C. K., Lee, H. D., Mayer, K. J., Sauer, J. S., Lee, C., Dorce, G., Malfatti, F., Stone, E. A., Cappa, C. D., Grassian, V. H., Prather, K. A. and Tivanski, A. V.: Size-Dependent Morphology, Composition, Phase State, and Water Uptake of Nascent Submicrometer Sea Spray Aerosols during a Phytoplankton Bloom, ACS Earth Sp. Chem., 6(1), 116–130, doi:10.1021/acsearthspacechem.1c00306, 2022a.

Zobrist, B., Marcolli, C., Pedernera, D. A. and Koop, T.: Do atmospheric aerosols form glasses?, Atmos. Chem. Phys., 8(17), 5221–5244, doi:10.5194/acp-8-5221-2008, 2008.

---

## Author Comment (AC2)

**ACP Response to RC2 for *Low Temperature Ice Nucleation of Sea Spray and Secondary Marine Aerosols under Cirrus Cloud Conditions***
Format: The reviewers' comments are quoted in italics
Line number in the response refers to the revised manuscript with tracked changes Quotation in red color stands for revised/added text in the revised manuscript
Responses in blue

*Patnaude et al. use the Marine Aerosol Reference Tank in combination with CFDC measurements to investigate the effect of atmospheric aging on the ice nucleation behavior of sea spray aerosols (SSA) under cirrus conditions. Simulated aging conditions led to the formation of a new particle mode of secondary marine aerosols (including also some contribution from VOCs outgassing from the MART) and induced some changes in the size distribution of the primary sea spray aerosol particles. These changes, however, had little effect on the observed ice nucleation behavior, which was still mainly governed by the inorganic salts as in primary sea spray aerosols.*

*The experiments were carefully conducted and analyzed and are therefore worth publishing. My main criticism is along the same lines as the other reviewer's: I would have liked to see more discussion of the nucleation mechanism and the phase state/hygroscopic behavior of the particles involved, which I have detailed in my comments below. I also highlighted a number of points where the discussion of earlier literature data must be improved.*

We thank the reviewer for their thoughtful comments and contributions to this manuscript. We will address both the major and minor below.

**Specific comments**

*P1, line 11 – 13: Please also mention here the drying step of the SSA to induce crystallization of the salt constituents prior to the CFDC measurements, as this is mandatory for the observation of heterogeneous freezing below 220 K.*

We have added a statement to make this clearer.

L11-L14: Polydisperse aerosol distributions were generated from a Marine Aerosol Reference Tank (MART) filled with 120 L of real or artificial seawater and were dried to very low relative humidity to crystallize the salt constituents of SSA prior to their subsequent freezing, which was measured using a Continuous Flow Diffusion Chamber (CFDC)."

*P1, line 20: "Thus, any SMA coatings on the pSSA are also unlikely to modify the ice nucleation behavior of pSSA." I cannot understand this conclusion. For other organic aerosols like secondary organic matter from the oxidation of terpenes and aromatic precursors investigated in the study by Kasparoglu et al. (2022), the pure SOM particles also required water saturation to freeze, but a coating layer of SOM could significantly de-activate efficient INPs like mineral dust at cirrus conditions. So, if a SMA coating does not change the ice nucleation behavior of pSSA, in my opinion two conditions have to be met: 1) the coating does not inhibit the crystallization of the inorganic salts in the drying step before entering the CFDC, and 2) water diffusion through the potential organic-rich coating layer around the salt crystals is also not significantly hindered.*

We agree your conditions would have to be met for the ice nucleation of the pSSA. We have added a sentence in the Abstract and Conclusions to address this comment.

L22-L24: "Thus, any SMA coatings on the pSSA would only decrease the ice nucleation behavior of pSSA if the organic components were able to significantly delay water uptake of the inorganic salts, and apparently this was not the case."

L798-801: "… the addition of SMA coatings to the pSSA and/or alteration of organic components of the pSSA did not alter the crystallization behaviors, nor did they hinder the water uptake by the inorganic salts. Nevertheless, the inclusion of organic material on the pSSA would likely only hinder the heterogeneous nucleation ability that remains dominated by the crystalline salts."

*P2, line 38 – 40: I found it somewhat confusing that the authors here also include the process of immersion freezing under mixed-phase cloud conditions, while in line 28 they specifically address ice formation at < 235 K. I would therefore also focus the description here on immersion freezing at cirrus condition, meaning that an INP initiates freezing from inside a sufficiently dilute aqueous solution droplet below the homogeneous freezing conditions for the aqueous layer.*

To alleviate any confusion, we have removed the second half of this sentence regarding immersion freezing under mixed-phase cloud conditions (0 to -38 ºC) and modified the first half to make this clearer.

L53-L55: "The immersion freezing mechanism under cirrus cloud conditions occurs when an INP initiates freezing from within a sufficiently dilute aqueous solution droplet at relative humidity lower than that required for homogeneous freezing conditions at a given temperature."

*P2, line 51: Please cite some sources discussing this "disagreement" on the contributions of black carbon and SOA.*

We have changed "disagreement" to "uncertainty" and added some additional statements following this.

L64-L71: "While black carbon (BC) and secondary organic aerosols (SOA) are found in abundance in the upper troposphere (Froyd et al., 2010; Schill et al., 2020; Schwarz et al., 2017) and appear to dominate IRs in cirrus contrails (Petzold et al., 1998), there is uncertainty in their ability to act as effective cirrus cloud INPs. For example, Mahrt et al. (2020) showed some enhanced ice nucleating ability of BC particles in cirrus clouds after cloud processing, but freshly emitted BC particles were ineffective. Another study found that BC particles were only effective at initiating heterogenous nucleation at cirrus temperatures if they were > 100 nm and contained surface pores to freeze via PCF (Mahrt et al., 2018). While some SOA have shown the ability to freeze heterogeneously at very low active fractions (Wolf et al., 2020a) it is generally assumed that the majority of SOA freeze via homogeneous nucleation."

*P2, line 53: "fractions too low to be meaningful" I find this statement too simplistic. There are other studies, such as the modeling work from Beer et al. (2022), which states in the abstract:*

*"On the other hand, crystalline ammonium sulfate often shows large INP concentrations, has the potential to influence ice nucleation in cirrus clouds, and should be taken into account in future model applications."*

*Beer, C. G., Hendricks, J., and Righi, M.: A global climatology of ice-nucleating particles under cirrus conditions derived from model simulations with MADE3 in EMAC, Atmos. Chem. Phys., 22, 15887– 15907, https://doi.org/10.5194/acp-22-15887-2022, 2022.*

Thank you for this suggestion. We have modified this statement to include the conclusions from Beer et al., (2022).

L71-L74: "Ammonium sulfate is abundant in the atmosphere and may also act in solid form as a cirrus INP, and although most studies have shown sulfate to freeze via heterogeneous nucleation at low active fractions (Abbatt et al., 2006; Baustian et al., 2010; Wise et al., 2010), their abundance may contribute meaningful impacts on cirrus cloud formation (Beer et al., 2022)."

*P3, line 77ff: I strongly suggest describing earlier literature results for pure SSA particles in more detail, e.g. with respect to their phase state and hygroscopic behavior, their strongly temperature-dependent ice nucleation behavior under cirrus conditions and the mode of ice nucleation. It is also useful to mention the efflorescence step required to induce crystallization of the particles. In addition, there are a number of studies suggesting that some salt components already dissolve at low humidity, which means that the previously dried SSA particles under ice-saturated conditions are already an internally mixed solid-liquid particle, i.e. with a brine layer around the undissolved NaCl core:*

Tang, I. N., Tridico, A. C., and Fung, K. H.: Thermodynamic and optical properties of sea salt aerosols, J. Geophys. Res. (Atmos.), 102, 23269-23275, 1997.

Schill, G. P., and Tolbert, M. A.: Heterogeneous Ice Nucleation on Simulated Sea-Spray Aerosol Using Raman Microscopy, J. Phys. Chem. C, 118, 29234-29241, 2014.

Wagner, R., Kaufmann, J., Möhler, O., Saathoff, H., Schnaiter, M., Ullrich, R., and Leisner, T.: Heterogeneous Ice Nucleation Ability of NaCl and Sea Salt Aerosol Particles at Cirrus Temperatures, J. Geophys. Res. (Atmos.), 123, 2841-2860, 2018.

*Even under cirrus conditions, but at temperatures above 220 K, the SSA particles "only" fully deliquesce and nucleate ice homogeneously. Heterogeneous ice formation is only observed below 220 K, and given that the particles are in an internally-mixed solid-liquid state, one could also speculate that the nucleation mode is immersion freezing.*

*Most of these aspects should then of course also be taken up when discussing the results, but I think it is very useful to give some more details on the IN behavior of pSSA already here in the introduction.*

We appreciate your suggestions regarding a more in-depth discussion regarding the heterogeneous freezing pathway of SSA at these temperatures and how the phase state of the particles may dictate the freezing pathway. We have added more discussion on this topic and have also included more discussion in the Conclusions section (see below).

L117-L129: "In Patnaude et al. (2021) we posited that our ice nucleation results for simpler model SSA were a result of the competition between full deliquescence and immersion freezing. Those freezing results at temperatures > 220 K were characteristic of homogeneous freezing, in agreement with other previous studies (Schill and Tolbert, 2014; Wagner et al., 2018), which was likely due to the full deliquescence of the salt components around ~75 % $RH_w$, as defined by Tang and Munkelwitz, (1993) that occurred prior to the onset of immersion freezing. Below 220 K, SSA particles have been found to initiate heterogeneous nucleation at/near the range of deliquescence relative humidities (DRHs) for NaCl and SSA. Both Schill and Tolbert, (2014) and Wagner et al. (2018) concluded that at temperatures below 220 K, the phase state of their SSA particles was characterized by a solid core with a brine layer shell, and speculated that heterogeneous freezing proceeded via the immersion pathway. Patnaude et al. (2021) further discussed the reason for the transition between homogeneous and heterogeneous pathways around 220 K, which also happened to occur at the intersection of the DRH and PCF freezing parameterization from Marcolli et al. (2014) for a specific pore size. However, due to the lower DRHs for the additional salt components such as magnesium and calcium (Tang et al., 1997) the PCF pathway is less likely. The

exact heterogeneous freezing mechanism of SSA and cause of the transition between freezing pathways remains unresolved."

*With regard to the potential change in the ice nucleation behavior of pure SSA when it contains a higher fraction of organic material, it is also useful to discuss the temperature ranges above 220 K and below 220 K separately. Above 220 K, the organic components could provoke an additional immersion freezing mode, while purely inorganic SSA only freezes homogeneously. Below 220 K, the question is whether and how the organic components modify the ice nucleation behavior of purely inorganic SSA. Some of this will be discussed later, but for the benefit of the reader, the important questions could be addressed already here. Similarly, you start the second paragraph of your conclusions (P9, line 341) by mentioning some open questions raised by Wagner et al. (2021). I recommend highlighting these open questions already in the introduction and describing how you intended to address them with your study.*

We have added a few additional statements on the possibility of organic particles to freeze above/below 220 K compared to the salt components of SSA.

L135-L146: "Additionally, Wagner et al. (2021) speculated on whether organic rich particles present in seawater would be aerosolized with a more realistic generation method and thus have the potential to act as heterogenous INPs. The inclusion of additional organic components in primary SSA may have implications for ice nucleation both below and above 220 K. Clearly, organics from seawater that are contained in primary SSA, and/or those added to the aerosol through atmospheric processing of SSA, are responsible for a major proportion of the ice nucleation activity of SSA in the mixed-phase cloud regime (McCluskey et al., 2018a), and a question has remained as to how effective these organics are as INPs below 238 K. Organic aerosols have been shown to form a glassy state under cirrus cloud conditions (Berkemeier et al., 2014; Ignatius et al., 2016; Knopf et al., 2018; Koop et al., 2011; Murray et al., 2010) and depending on the glass transition temperature and DRH, these particles may have heterogeneous ice nucleating ability at temperatures > 220 K, where the salt components of SSA would have already dissolved. It is unknown how organic rich particles would compete with the salt components of SSA below 220 K, but based on the results in Kasparoglu et al. (2022), the contribution of glassy SOA may be relatively minor."

*P5, line 163ff: This paragraph contains a detailed description of the size distribution measurements, which in itself is of course very justified, but I sometimes found it difficult to follow the lines of thought in the text. For example, in line 167/168 you specifically mention the different diameters at which the aSSA+SMA distributions converge to the pSSA distributions for ASW and SW. This is then immediately followed by the statement "This is consistent …" in line 168. I had expected an explanation here for this different "convergence" behavior in this sentence, but unless I misunderstood, this is given later in line 171. Instead, the sentence with "This is consistent …", if I interpreted it correctly, refers only to the fact that the aSSA+SMA distributions were dominated by secondary particle formation. So you might consider rearranging the text a little.*

Thank you for the suggestion. We have added significant discussion to this section and have moved it to a new results section 3.1. This involves more detailed description of each of the three aerosol types and discussion on the composition and phase state, specifically for the aSSA+SMA particle type. We hope it is now clearer.

L548-L567: "Although there were higher number concentrations of aSSA+SMA between 200 nm and 1 μm, there was not a discernable shift towards larger sizes, which would be expected after condensation from the gas phase onto the pSSA. In addition, the low concentrations of pSSA would limit condensation of organics in the OFR and would favor new particle formation. The estimated condensation sink timescales were calculated and shown in the supplemental material, indicating that the condensation sink

timescale for the nucleation mode aerosols (SMA-only) was ~3 minutes compared to ~11 minutes for the pSSA. Therefore, it is likely the nucleation mode particles scavenged the majority of the condensable material in the OFR and minimized condensation onto the pSSA. However, prior OFR studies for similar pSSA loadings indicated modest increases in organic volume fractions in pSSA as detected by atomic force microscopy (AFM) following similar oxidation exposures using the same OFR (Kaluarachchi et al. 2022b). These changes were accompanied by modifications in particle phase state in the water subsaturated regime, as well as hygroscopicity, for submicron pSSA. DeMott et al. (2023) discussed how similar OFR studies on laboratory generated pSSA led to apparent changes in organic functionalization, at least as determined by AFM for submicron particles. Raman spectroscopy did not detect functional changes following oxidation in that study, but this was inferred to be a consequence of the emphasis on the 1 μm and larger particle regime for the Raman spectroscopic studies, for which organic volume fractions are already quite small and signal to noise becomes an issue. Finally, DeMott et al. (2023) noted that OFR processing of pSSA led to a decrease in INP concentrations by a factor of a few times in the temperature regime >243 K, where heterogeneous ice nucleation has been shown to be initiated by the organic components in very small fractions of pSSA. It has been unknown if this alteration of ice nucleation at higher temperatures translates to impacts at cirrus temperatures. Further, it is also not known whether changes induced by oxidization of the organics present in the pSSA, or the changes in ice nucleating activity due to organics added via condensation or functional alteration might affect the heterogeneous nucleation process that was inferred to be stimulated by crystalline salts at temperatures below 220 K (Patnaude et al., 2021)."

L568-L577: "Both pSSA size distributions showed peaks around 100 nm, in agreement with other studies that generated SSA in laboratory settings (Collins et al., 2014; Patnaude et al., 2021; Quinn et al., 2015). However, there was a slight secondary mode at ~800 nm that only occurred during the real seawater experiments (Figure 4b). These two aerosol modes at 100 nm and 800 nm likely represent particles generated from film and jet drops, respectively, consistent with a previous laboratory study that generated aerosol particles from real seawater (Hill et al., 2023). Prather et al. (2013) showed that the fraction of biological particles increased for particles > 1 μm, and can be up to ~20 % of particles > 2 μm. Therefore, this apparent mode of particles ~800 nm may represent additional aerosolized biologic material such as bacteria, gels and viruses, that may be enriched with organic particles (Hill et al., 2023), would not exist in the ASW, and may explain lower concentrations in the aSSA+SMA generated from ASW compared to SW between 200 nm and 1 μm. The differences in the ~800 nm mode could also be due to modifications in droplet generation mechanisms, through small differences in temperature, RH, and surfactant content (Stokes et al., 2013)."

*As another example, you mention in line 182 that the blank test with DI water "also" led to secondary particle formation. But in fact, in a certain diameter range between 80 to 200 nm, it even led to the highest formed particle number in all experiments (Fig. 2a). This could easily give the impression that new particle formation in the OFR is exclusively governed by impurities/organics from the acrylic outgassing and that direct gas-phase emission from SW and ASW (do you expect a lot of organics from ASW?) is of minor importance. Only later do you give the information "as a final note" (line 188) that the blank test might have been affected by the higher temperature compared to SW and ASW – I think this information should also be given earlier in the discussion. You have also included Fig. S2 in this context to show the change in size distribution with experiment hours (and a concomitant change in temperature). Unfortunately, the y-axis scale in Fig. S2a is given on a linear scale, plotting it with a log-scale as in Fig. 2a would allow a better comparison of the two panels.*

We have also rearranged the portion of the paragraph focusing on the formation of secondary aerosols, including bringing the statement on the effects of temperatures towards the start of this discussion. We have also changed Figure S2a to be on a log-scale.

L578-L586: "For the blank (background) tests, in which the MART was filled with only DI water and the OFR was turned on but MART plunging turned off (black line), it was found that there was also secondary particle formation, likely a result of VOCs emitted from the acrylic material of the MART walls. It is likely that the outgassing of acrylic tank materials and subsequent aerosol formation from those compounds in the OFR occurred in all oxidation experiments. The outgassing may have varied with water temperature, which increased throughout each experimental day, and could contribute to some of the differences observed in the aSSA+SMA and SMA-only size distributions. For example, during the pSSA and aSSA+SMA experiments, particle concentrations below 100 nm were higher later in the day due to the warming of the water with time (Figure S2). During the blank test the water was not pre-cooled as in the SW and ASW experiments due to time constraints, which likely led to higher VOC emissions and may explain the high secondary particle concentrations observed during the blank measurements."

[Figure]

**Figure S2.** Size distributions of (a) aSSA+SMA and (b) pSSA color-coded by hours of each experiment.

*P6, line 195: There are a couple of further studies investigating the temperature-dependent efflorescence behavior of NaCl, apart from Koop et al., e.g.:*

Bartels-Rausch, T., Kong, X., Orlando, F., Artiglia, L., Waldner, A., Huthwelker, T., and Ammann, M.: Interfacial supercooling and the precipitation of hydrohalite in frozen NaCl solutions as seen by X-ray absorption spectroscopy, The Cryosphere, 15, 2001–2020, https://doi.org/10.5194/tc-15-2001-2021, 2021.

Peckhaus, A., Kiselev, A., Wagner, R., Duft, D., and Leisner, T.: Temperature-dependent formation of NaCl dihydrate in levitated NaCl and sea salt aerosol particles, J. Chem. Phys., 145, 244503, 2016.

Wagner, R., Möhler, O., and Schnaiter, M.: Infrared Optical Constants of Crystalline Sodium Chloride Dihydrate: Application to Study the Crystallization of Aqueous Sodium Chloride Solution Droplets at Low Temperatures, J. Phys. Chem. A, 116, 8557-8571, 2012.

Thank you for the suggestions of additional references. We have included these at the end of this sentence.

L243-L247: "Previous studies have found that the phase state of NaCl after efflorescence depends on the temperature at which efflorescence occurred, with anhydrous NaCl observed for efflorescence temperatures above 273 K (Bartels-Rausch et al., 2021; Koop et al., 2000b; Peckhaus et al., 2016; Wagner et al., 2012). Since the majority of aerosol drying, including the efflorescence of the salt components of pSSA and aSSA+SMA, occurred at room temperature (Figure 2), it is believed the salts would be in the anhydrous form."

*So the statement "phase state and morphology are less well understood at low temperature" should also be revised.*

We have revised this sentence for clarity that what is uncertain in the phase state of any additional organic particles or coatings on or functional alteration of the pSSA.

L247-L249: "However, the phase state and morphology of co-emitted organic particles or coatings on the pSSA and aSSA+SMA are less well understood at low temperatures, and the exact structure was uncertain in this study."

*P6, line 199: "porous glassy state": However, the formation of such porous glassy particles requires a special process called "atmospheric freeze-drying in ice clouds" by Adler et al. (2013). I do not think this is relevant to your study.*

Thank you. We removed that reference and modified the sentence.

L259-L260: "Aerosols that contain organic material may either enhance heterogeneous nucleation (Wilson et al., 2012) or have no effect (Kasparoglu et al., 2022)."

*P6, line 200-204: This behavior is not a contradiction, but can be explained by the different viscosity of the coating material at cirrus temperatures compared to mixed-phase cloud conditions, i.e., a more liquid-like behavior at higher temperatures, and more solid, glassy-like behavior at cirrus conditions, e.g. Fig. 5 in Charnawskas et al. (2017):*

Charnawskas, J. C., Alpert, P. A., Lambe, A. T., Berkemeier, T., O'Brien, R. E., Massoli, P., Onasch, T. B., Shiraiwa, M., Moffet, R. C., Gilles, M. K., Davidovits, P., Worsnop, D. R., and Knopf, D. A.:

Condensed-phase biogenic–anthropogenic interactions with implications for cold cloud formation, Faraday Discuss., 200, 165-194, 10.1039/C7FD00010C, 2017.

Thank you for the suggestion from Charnawskas et al. (2017). We have added a sentence to make the point that the temperature matters with regard to the effects of organic coatings on heterogeneous freezing.

L263-L267: "At warmer temperatures > 233 K, organic coatings may be in a less viscous or liquid-like phase state, while at colder temperature may become semi-solid or glassy (Charnawskas et al., 2017), which may explain the differences in ice nucleation behavior of particles with organic coatings in those studies. Specifically, the inclusion of glassy coatings on the pSSA or aSSA+SMA particles may inhibit water uptake and delay dissolution of the salt components until higher $RH_w$ is reached."

*P6, sect. 2.3:*
*I really liked this section, a very careful analysis of how to derive the total number of ice crystals. I think it would also be useful to describe the expected hygroscopic behavior of the pSSA - I have already mentioned this in my comment on P3, line 77ff (i.e., that one would expect a small water uptake already at low RH and the full deliquescence when finally the remaining NaCl fraction is dissolved). Have you also looked at the smaller OPC size channels in Fig. 3a to see if you can detect the full deliquescence step of the pSSA particles there, like e.g. in Figs. 2 & 3 of Kong et al. 2018:*

Kong, X. R., Wolf, M. J., Roesch, M., Thomson, E. S., Bartels-Rausch, T., Alpert, P. A., Ammann, M., Prisle, N. L., and Cziczo, D. J.: A continuous flow diffusion chamber study of sea salt particles acting as cloud nuclei: deliquescence and ice nucleation, Tellus B, 70, 10.1080/16000889.2018.1463806, 2018.

Thank you for these very good suggestions. We could not directly replicate Figure 2 from Kong et al., (2018) because the OPC in this study could not measure the smallest particles in the size distribution. But we did replicate their Figure 3 and have now included this in our supplemental material. Below is the new figure and our additional discussion in section 2.3. Unfortunately, we could not replicate this figure exactly because our RH scans at 233 K did not go below ~73 % $RH_w$ where the particles would already be deliquesced. Instead, we chose to use 213 K (-60 ºC) where we had a full range of RHs below 70 % and up to > 85 %.

L463-L470: "Figure S3 shows a timeline of the particle counts from lower OPC channels, similar to Figure 3 from Kong et al. (2018) but does not show the sharp step where deliquescence occurred. This may be due to several reasons, such as the lower vapor pressures at a colder temperature (213 K versus 233 K in Kong et al. (2018)), or that in this study we used a polydisperse aerosol distribution so smaller particles taking up water would shift into the same OPC channels that larger particles should be shifting out of toward larger sizes. However, both the pSSA and aSSA+SMA did show some decrease in the number of particles as the $RH_w$ increased above the deliquescence RH (~74 %) indicating more gradual water uptake and slower deliquescence transition at lower temperatures."

[Figure]

**Figure S3.** Time series of particle counts for four different OPC size channels for (a) pSSA and (b) aSSA+SMA at 213 K. The cyan line represents the $RH_w$ and the dashed red line represents the $RH_w$ where the particle counts begin to decrease indicating water uptake and particle growth.

*And what do you think is most likely the heterogeneous ice nucleation mode seen in Fig. 3b?*

As discussed in the Results section 3.2, we believe the most likely heterogeneous ice nucleation mechanism is immersion freezing. We do not believe this discussion should be included in this section. Hence, we added a sentence that the inferred mechanism of freezing will be discussed later in the manuscript.

L478: "The inferred mechanisms of freezing at these temperatures will be discussed below."

*P8, line 274: "were broadly consistent": For me, this is a rather insufficient comparison with earlier literature data. You should present this in more detail and include earlier data in Fig. 5.*

We do not believe it is necessary to include the earlier data as this is already a fairly busy figure. Our point was only that the time series showed the freezing mechanisms were similar depending on the temperatures. We have added more to this sentence to make it clearer.

L605-L607: "These results were consistent with previously reported freezing behaviors of NaCl particles at temperatures between 205 K and 235 K (Patnaude et al., 2021; Wagner et al., 2018) who showed time series of freezing indicative of heterogeneous freezing below ~218 K and homogeneous freezing above 218 K."

*P8, line 279: "procedure was modified": Yes, indeed, when I looked at Fig. 4b, I wondered about the changed shape of the SSi ramps. Could you explain a little more what the purpose of the changed procedure was?*

This figure has been modified based also on comments from other reviewers. The change in $RH_w$ ramp was a minor procedural change made to mitigate the inability of the optical particle counter to resolve, without counting errors, the very high concentrations of particles growing into its lower size range for the SMA experiments when the transition to homogeneous freezing was achieved very quickly. Therefore, we decreased $dRH_w/dt$ in these experiments so that the peak $RH_w$ of 102 % was approached more gradually and the ice active fraction could be resolved for high numbers of SMA. In the updated figure, the upper limit in $RH_w$ is the same.

[Figure]

**Figure 5.** Time series of CFDC scans for (a) pSSA and (b) SMA experiments generated from real seawater beginning at 204 K. The CFDC column temperature is represented by the red lines, and the blue lines represents the $RH_w$. The black and light blue markers indicate the numbers of total particles counted in the OPC and those that are considered ice crystals, respectively. The black dotted horizontal lines in the top panels of (a) and (b) represent water saturation.

*P8, line 292: You provide here a "fit" to your low-temperature ice nucleation data within the framework of the pore condensation and freezing (PCF) concept, but – as far as I can tell – never come back to this*

*point in the following discussion. As mentioned above, you should really elaborate on what you think the possible heterogeneous nucleation mechanism might be.*

See our response to your earlier comment above regarding the mechanisms for heterogeneous freezing pathways where we discuss PCF as a possible mechanism. We have also added more discussion on this as a possible freezing pathway in the Results section 3.2 and Conclusions.

L720-L734: "Similar to Patnaude et al. (2021), the pathway for heterogeneous nucleation and the transition between homogeneous and heterogeneous below 220 K remains unresolved. Deposition freezing remains an unlikely freezing pathway for pSSA and aSSA+SMA (at 1 % frozen fraction) at/near the DRH as water uptake of minor salt constituents would begin at lower RHs (Schill and Tolbert, 2014; Tang et al., 1997; Wagner et al., 2018). The competition between full deliquescence and immersion freezing or PCF to explain this transition in freezing pathways was also discussed in Patnaude et al. (2021) but bears repeating. While the 1% frozen fraction of both pSSA and aSSA+SMA occurred at near the PCF line, we do not believe this to be the likely freezing pathway for two reasons: 1) the PCF line shown in Figure 6 was approximated for an 11 nm pore size, and there is little evidence this is an realistic size for SSA and 2) the SSA particles would need to be fully dry to retain surface pores, which was not likely the case at 75 % RH$_w$ where 1 % frozen fraction occurred. At temperatures above ~220 K, particle deliquescence would occur at a lower RH$_w$ than the expected value for immersion freezing (see Patnaude et al., (2021) Figure 8), thus favoring full dissolution and homogeneous nucleation at much higher RHs. At lower temperatures (< 220 K), the particles may have developed a brine layer, which has been observed in two previous studies (Schill and Tolbert, 2014; Wagner et al., 2018), and ice nucleation may have proceeded via the immersion pathway. Therefore, the most likely heterogeneous freezing pathway for pSSA and aSSA+SMA was immersion freezing."

L766-L767: "Based on the similarities between freezing results in this study and Patnaude et al. (2021) and Wagner et al. (2018), the dominant heterogeneous freezing mechanism below 218 K was likely the immersion freezing pathway."

*P8, line 297: "range of uncertainties": Could you please add respective error bars on a couple of data points in Figs. 5/6 and Figs. S4/S5?*

Because the uncertainty is in the calculation of SSw, it is not a random error, but a bound on the accuracy of the measured SSw. This is the same for all points shown on the graph, such that if our measurement is inaccurate, they would all shift together. Including it as an error bar in the figures is misleading because it leads to the interpretation that there might be that amount of random difference between points in the plots, which is not accurate.

A sentence has been added after the discussed line that reads:

L631-L632: "It should be noted that this is not a random uncertainty, that if there is error in the calculated RH$_w$ it will be the same for all measured points"

*P9, line 318/319: You might be a bit more specific here and say that in the case of SSA it is the "full" deliquescence RH (because these multi-component salts show a gradual deliquescence behavior, see Tang et al. (1997)).*

L712-L714: "Additionally, for both water types, the onset of heterogeneous freezing occurred near the estimated deliquescence RHs of NaCl and the full deliquescence of SSA (blue shaded region), similar to the results in previous studies (Patnaude et al., 2021; Wagner et al., 2018, 2021)."

*P9, line 341ff: As noted above, the mention of these open questions comes rather abruptly. You should also address these questions in the introduction and explain how you conducted/planned your experiments to answer these questions.*

Good point, we addressed this by adding a sentence in the Introduction. See response to an earlier comment about this same topic.

*P10, line 353ff: The final sentence of your abstract reads as a very general conclusion that primary SSA remain efficient INPs even after atmospheric aging. The aging conditions in your experiments predominantly led to the formation of new, less ice-active SMA particles, and only to a lesser extent to a change in the size distribution (and thus composition) of the pSSA particles. Can you rule out atmospheric scenarios (different aging conditions) where a larger amount of condensable material is transferred to the pSSA particles, so that they could form a thicker coating layer of organics, which could then affect the efflorescence behavior and possibly hinder water diffusion?*

We agree; although the conditions in the OFR were designed to mimic the marine boundary layer, in the MBL the vapor would not likely supersaturate in the same way and pSSA could scavenge more of the low-volatility organics produced via oxidation. It may be possible that the freezing behavior of pSSA may be different if we had used a more realistic oxidation method, or had fully effloresced the particles prior to oxidation. Therefore, we modified the final sentence of the Abstract such that these results only apply to this study, and we also added a few sentences in the Conclusions section.

L24-L27: "Results from this study demonstrate the ability of lofted primary sea spray particles to remain an effective ice nucleator at cirrus temperatures, even after atmospheric aging has occurred over a period of days in the marine boundary layer prior to lofting. We were not able to address aging processes under upper tropospheric conditions."

L801-L804: "Future studies using more realistic or at least slower atmospheric aging processes, such as what may occur in a typical smog chamber, may better represent the possible secondary organic coatings on pSSA. Additionally, oxidation of pSSA at cold temperatures (< 233 K) and after efflorescence may produce different freezing results than this study due to oxidation under different phase states and morphologies."

**Technical corrections**

*P1, line 14: Maybe better state: "homogeneous freezing conditions (xx -xx %RH) have to be reached to freeze 1% of the particles.*

L14-L16: "Results show that for both primary SSA (pSSA), and the aged SSA and SMA (aSSA+SMA) at temperatures > 220 K, homogeneous conditions (92 – 97 % relative humidity with respect to water (RH$_w$)) were required to freeze 1 % of the particles."

*P2, line 42: "pore condensation and freezing"*

L56 – L58: "However, in recent years this mechanism has been debated and some consider the freezing event to be a result of pore condensation and freezing (PCF), where water vapor condenses into pores or cracks on the surface of an INP (David et al., 2019; Marcolli, 2014)."

*P2, line 65: Delete the point after "mass" and replace the comma by a point after the citation.*

L83-85: "SMA have been shown to be comprised primarily of non-sea-salt (nSS) sulfates and other secondary organic species, produced primarily from the oxidation of DMS, which can dominate the total submicron aerosol mass (Fitzgerald, 1991; O'Dowd et al., 2004)."

*P4, line 132: A very nested sentence, could you please rephrase it?*

L192-L193: "The aerosol sample line split in two after exiting the MART with one sample line sent through two silica gel diffusion dryers to the upstream set of sizing instruments to confirm the nascent SSA number size distributions produced by the MART."

*P5, line 161: 1.9 g cm-3*

L289-L239: "The APS measurements were converted from aerodynamic to geometric diameter using a dynamic shape factor and particle density of SSA of 1.05 and 1.9 g $cm^{-3}$,…"

*P5, line 163: You might highlight here that Fig. 2 now addresses only the downstream size distribution measurements.*

L481-L482: "Data from the SMPS and APS downstream of the OFR were merged into a complete size distribution, as shown in Figure 4 for all experiments."

*P7, line 246 and 261: What means "IS" filters?*

IS was short for ice spectrometer, which we defined in section 2.2. See response below for the removal of this section.

*P7, line 262: What do you conclude from the agreement between real SW and ASW? I am rather surprised by this.*

We cannot fully describe the reasons for the similarities between the INP spectra from SW and ASW. It may be a result of very low INP concentrations from the SW (low at this time of year, based on previous samples) and not collecting sufficient air volume on the filters to show a more significant difference between the ASW and SW. There may also be inorganic INP contamination of the ASW during manufacturing. We have decided to remove this figure to alleviate the confusion as the INP concentrations in the mixed-phase regime have little to do with the low temperature ice nucleation, which is dominated by the inorganic salts.

*P8, line 271: Maybe better: "beginning at 218 K and above"*

L598-L600: "The final three scans beginning ~218 K and above, showed a more modest increase in ice particle counts until the $RH_w$ reached nearly 100 %, resulting in a very sharp increase in ice particle counts."

*P9, line 338: One could rephrase this a bit and say that heterogeneous ice formation with activated fractions above 1% only occurred at temperatures of 218 K or below.*

L761-L764: "This was illustrated by the striking similarities in ice nucleation results between the pSSA from this study and the results from Patnaude et al., (2021), which both determined heterogeneous ice formation with activated fractions above 1 % only occurred at temperatures of 218 K or below."

*P10, line 380: All co-authors contributed*

L829: "All co-authors contributed to reviewing and editing of the manuscript."

*Caption of Fig. S5 in the supplement: Check sentence in line 29, probably delete the first "difference"*

We have deleted the first difference.

**New References:**

DeMott, P. J., Hill, T. C. J., Moore, K. A., Perkins, R. J., Mael, L. E., Busse, H. L., Lee, H., Kaluarachchi, C. P., Mayer, K. J., Sauer, J. S., Mitts, B. A., Tivanski, A. V, Grassian, V. H., Cappa, C. D., Bertram, T. H. and Prather, K. A.: Atmospheric oxidation impact on sea spray produced ice nucleating particles, Environ. Sci. Atmos., doi:10.1039/d3ea00060e, 2023.

Hill, T. C. J., Malfatti, F., McCluskey, C. S., Schill, G. P., Santander, M. V., Moore, K. A., Rauker, A. M., Perkins, R. J., Celussi, M., Levin, E. J. T., Suski, K. J., Cornwell, G. C., Lee, C., Del Negro, P., Kreidenweis, S. M., Prather, K. A. and DeMott, P. J.: Resolving the controls over the production and emission of ice-nucleating particles in sea spray, Environ. Sci. Atmos., 3(6), 970–990, doi:10.1039/D2EA00154C, 2023.

Kaluarachchi, C. P., Or, V. W., Lan, Y., Hasenecz, E. S., Kim, D., Madawala, C. K., Dorcé, G. P., Mayer, K. J., Sauer, J. S., Lee, C., Cappa, C. D., Bertram, T. H., Stone, E. A., Prather, K. A., Grassian, V. H. and Tivanski, A. V.: Effects of Atmospheric Aging Processes on Nascent Sea Spray Aerosol Physicochemical Properties, ACS Earth Sp. Chem., 6(11), 2732–2744, doi:10.1021/acsearthspacechem.2c00258, 2022b

Mahrt, F., Marcolli, C., David, R. O., Grönquist, P., Barthazy Meier, E. J., Lohmann, U. and Kanji, Z. A.: Ice nucleation abilities of soot particles determined with the Horizontal Ice Nucleation Chamber, Atmos. Chem. Phys., 18(18), 13363–13392, doi:10.5194/acp-18-13363-2018, 2018.

Mahrt, F., Kilchhofer, K., Marcolli, C., Grönquist, P., David, R. O., Rösch, M., Lohmann, U. and Kanji, Z. A.: The Impact of Cloud Processing on the Ice Nucleation Abilities of Soot Particles at Cirrus Temperatures, J. Geophys. Res. Atmos., 125(3), 1–23, doi:10.1029/2019JD030922, 2020.

---

## Author Comment (AC3)

**ACP Response to RC3 for *Low Temperature Ice Nucleation of Sea Spray and Secondary Marine Aerosols under Cirrus Cloud Conditions***

Format: The reviewers' comments are quoted in italics
Line number in the response refers to the revised manuscript with tracked changes Quotation in red color stands for revised/added text in the revised manuscript
Responses in blue

*Patnaude et al. characterized the formation of ice nucleating particles in primary sea spray aerosols (pSSA), secondary marine aerosols (SMA), and a mixture of oxidized SSA and SMA generated in a marine aerosol reference tank and exposed to hydroxyl radicals in an oxidation flow reactor. They concluded that heterogenous ice nucleation occurred with pSSA, homogenous ice nucleation occurred with SMA, and that the INP formation potentials of aSSA + SMA and pSSA were similar.*

**Comments**

*In the abstract (L16-L18), based on results presented later in the paper, the authors state: "Similarities between freezing behaviors of the pSSA and aSSA+SMA at all temperatures suggest atmospheric aging has little effect on the heterogeneous freezing behavior of SSA at these cirrus temperatures and remains dominated by the crystalline salts." I think this conclusion is flawed the way it is presented. It might be applicable to the specific experimental conditions used in this manuscript, but it seems unlikely to me that it can be generalized to other SSA and SMA. First, I couldn't find any characterization of organic content in the pSSA generated here. It seems plausible to me that the MART-generated pSSA could be relatively inert simply because it has negligible organic content, whereas pSSA with higher organic content might be more reactive towards OH and therefore experience greater changes in IN formation potential. Second, no concrete evidence of SMA coating the pSSA or aSSA was provided - as the authors pointed out, the specific conditions that were used strongly favored homogenous nucleation of SMA over condensation of SMA onto SSA. They note that "the higher concentration of aSSA+SMA compared to pSSA from natural SW between 200 nm and 1 μm (Figure 2b) suggests some modification of the pSSA" but there is no way to prove that this is associated with gas-phase condensation over the other possible explanations that were provided (L172-L174). If it were due to condensation, the aSSA+SMA size distribution should be shifted towards larger particle sizes rather than a higher concentration of the same size particles. I interpreted Figure 2b to suggest that there is negligible condensed SMA on the SSA. Third, because this paper demonstrates that the IN formation efficiency of SMA is clearly lower than that of SSA, a mixed aSSA+SMA particle should at some threshold SMA:SSA ratio have lower IN formation potential than pure SSA particles. I don't know what that ratio is, but it doesn't seem to me that it was reached here. Overall, it seems valid to conclude that SMA doesn't form INP as efficiently as SSA, and that the IN formation efficiency of these pSSA didn't change appreciably following OH exposure, but I don't think any sort of definitive conclusions can be made regarding the IN formation efficiency of mixed SSA/SMA particles or of other pSSA particles that might contain more organics.*

We thank the reviewer for the thorough read and for the many helpful suggestions, which we have carefully considered. Below, we respond to the various comments.

With regard to the comments on the lack of characterization of organic content in the pSSA generated: while it is true that we did not independently characterize the organic content of our natural seawater sample, it was obtained from a location (Scripps Pier) from which similar samples have been extensively studied in prior lab experiments. Therefore, we have added a sentence and reference to note there was high confidence that the organic content of the pSSA was fairly representative on the basis of a number of prior studies generating SSA from seawater in a MART.

L570-L576: "These two aerosol modes at 100 nm and 800 nm likely represent particles generated from film and jet drops, respectively, consistent with a previous laboratory study that generated aerosol particles from real seawater (Hill et al., 2023). Prather et al., (2013) showed that the fraction of biological particles increased for particles > 1 µm, and can be up to ~20 % of particles > 2 µm. Therefore, this apparent mode of particles ~800 nm may represent additional aerosolized biologic material such as bacteria, gels, and viruses, that may be enriched with organic particles (Hill et al., 2023), would not exist in the ASW, and may explain lower concentrations in the aSSA+SMA generated from ASW compared to SW between 200 nm and 1 µm."

L249-L258: "It is assumed that particles generated from SW contain high fractions of insoluble organic particles or salt particles mixed with organic carbon below 500 nm, similar to previous lab-generated SSA (Prather et al. 2013; Bertram et al. 2018; Kaluarachchi et al. 2022a). In addition, DeMott et al. (2023) showed particle morphologies of laboratory-generated SSA that include organic coatings in the submicron size range, both before and after similar use of an OFR for oxidation studies. The MART was demonstrated to produce substantially similar size distributions and SSA compositions compared to a more natural wave breaking process for bubble bursting (Prather et al., 2013; Stokes et al., 2013), so no bias in organic content in comparison to natural SSA production is expected. During cooling, the $RH_w$ would be low enough such that organic particles or coatings may form a glassy state (Ignatius et al., 2016; Knopf et al., 2018; Koop et al., 2011), as shown by the shaded green region in Figure 2, which represents the glass transition conditions for sucrose (Zobrist et al., 2008)."

We have also added a few sentences to address your concerns about the lack of organic condensation onto the pSSA.

L548-L567: "Although there were higher number concentrations of aSSA+SMA between 200 nm and 1 µm, there was not a discernable shift towards larger sizes, which would be expected after condensation from the gas phase onto the pSSA. In addition, the low concentrations of pSSA would limit condensation of organics in the OFR and would favor new particle formation. The estimated condensation sink timescales were calculated and shown in the supplemental material, indicating that the condensation sink timescale for the nucleation mode aerosols (SMA-only) was ~3 minutes compared to ~11 minutes for the pSSA. Therefore, it is likely the nucleation mode particles scavenged the majority of the condensable material in the OFR and minimized condensation onto the pSSA. However, prior OFR studies for similar pSSA loadings indicated modest increases in organic volume fractions in pSSA as detected by atomic force microscopy (AFM) following similar oxidation exposures using the same OFR (Kaluarachchi et al. 2022b). These changes were accompanied by modifications in particle phase state in the water subsaturated regime, as well as hygroscopicity, for submicron pSSA. DeMott et al. (2023) discussed how similar OFR studies on laboratory generated pSSA led to apparent changes in organic functionalization, at least as determined by AFM for submicron particles. Raman spectroscopy did not detect functional changes following oxidation in that study, but this was inferred to be a consequence of the emphasis on the 1 µm and larger particle regime for the Raman spectroscopic studies, for which organic volume fractions are already quite small and signal to noise becomes an issue. Finally, DeMott et al. (2023) noted that OFR processing of pSSA led to a decrease in INP concentrations by a factor of a few times in the temperature regime >243 K, where heterogeneous ice nucleation has been shown to be initiated by the organic components in very small fractions of pSSA. It has been unknown if this alteration of ice nucleation at higher temperatures translates to impacts at cirrus temperatures. Further, it is also not known whether changes induced by oxidization of the organics present in the pSSA, or the changes in ice nucleating activity due to organics added via condensation or functional alteration might affect the heterogeneous nucleation process that was inferred to be stimulated by crystalline salts at temperatures below 220 K (Patnaude et al., 2021)."

Finally, we have modified our conclusions to note that although the SMA had very inefficient ice nucleating behavior, we also infer that any condensational contributions to the pSSA did not alter the ice nucleating behavior for a number of different possible reasons. We added a few more clear statements in the Conclusions and also modified the Abstract.

L17-L24: "Similarities between freezing behaviors of the pSSA and aSSA+SMA at all temperatures suggest that the contributions of condensed organics onto the pSSA or alteration of functional groups in pSSA via atmospheric aging did not hinder the major heterogeneous ice nucleation process at these cirrus temperatures, which have previously been shown to be dominated by the crystalline salts. Occurrence of 1 % frozen fraction of SMA, generated in the absence of primary SSA, was observed at/near water saturation below 220 K, suggesting it is not an effective INP at cirrus temperatures, similar to findings in the literature for other organic aerosols. Thus, any SMA coatings on the pSSA would only decrease the ice nucleation behavior of pSSA if the organic components were able to significantly delay water uptake of the inorganic salts, and apparently this was not the case."

L782-L804: "There are a few possible reasons for the similar freezing behaviors between these two particle types: 1) the newly formed particles scavenged the majority of condensable material produced in the OFR and only modest amounts of secondary organics condensed to the pSSA, 2) the air stream remained humidified when entering the OFR and the pSSA particles were thus likely in an aqueous state. Whether oxidation of their organic content could proceed in the wetted or partially wetted state, and whether changes to the pSSA organic content would be similar to that observed when processed in a dry crystalline state is unknown, or 3) the addition of SMA coatings to the pSSA and/or alteration of organic components of the pSSA did not alter the crystallization behaviors, nor did they hinder the water uptake by the inorganic salts. Nevertheless, the inclusion of organic material on the pSSA would likely only hinder the heterogeneous nucleation ability that remains dominated by the crystalline salts. Future studies using more realistic or at least slower atmospheric aging processes, such as what may occur in a typical smog chamber, may better represent the possible organic coatings on pSSA. Additionally, oxidation of pSSA at cold temperatures (< 233 K) and after efflorescence may produce different freezing results than this study due to oxidation under different phase states and morphologies."

*L114 - If it's important enough to mention that the "grow light was set to a realistic PAR", please state what the radiation flux was.*

L173-L175: "The grow light was set to a realistic PAR (photosynthetically active radiation) quantity of ~175 $\mu$mol m$^{-2}$ to keep the microorganisms active and not stimulate a bloom."

*L124 – You might consider modifying the section title to something like "pSSA, aSSA + SMA, and SMA generation and characterization"*

Thank you for your suggestion, we have instead changed the section title to "Aerosol generation and characteristics".

*L133 - There are a few commercial and many home-built OFR designs; based on ensuing details in the following sentences and the information provided in Mayer et al. (2020), this is most likely an Aerodyne Potential Aerosol Mass OFR, but this detail should be clarified in the text.*

L153-L155: "Using an Aerodyne Potential Aerosol Mass OFR, we examined the impact of atmospheric oxidation on freezing behavior of primary SSA at cirrus temperatures."

*L135 – The text states: "The OFR generates [...] O$_3$ and OH radicals [...] using two UV lamps at wavelengths $\lambda = 254$ nm and $\lambda = 184$ nm [...] at a 90:10 ratio, with 90 % intensity from $\lambda = 254$ and 10 %*

*from λ =184 nm." This is not correct. First, the secondary Hg emission line in ozone-producing UVC lamps is centered at ~185 nm, not 184 nm. Second, and much more important, the ratio of 254 and 185 nm radiation fluxes is not 90:10. As far as I can tell, the OFR was operated with lamp type "GPH436T5L/VH/4P 90/10", which has doped:fused quartz fractions of 90% and 10%, but emits 185 nm radiation at approximately 0.6% of the intensity of the 254 nm radiation (Rowe et al., 2020).*

Thank you for pointing out 185 nm, that was a typo and we have changed the text to indicate185 nm and the ratio of lamp intensities as 0.6%.

L196-L199: "The OFR generates high concentrations of $O_3$ and OH radicals to oxidize particles using two UV lamps at wavelengths λ = 254 nm and λ =185 nm (Mayer et al., 2020). The UV lamp type used in this study was GPH436T5L/VH/4P 90/10, which emits radiation at the 185 nm wavelength at 0.6 % of the intensity of the 254 nm wavelength (Rowe et al., 2020)."

*L137 – 7 ppm $O_3$ does not translate to 4-6 days of equivalent atmospheric OH oxidation. Please specify the corresponding OH exposure, or range of OH exposure(s), and how they were measured or calculated, along with the ambient OH concentration that was assumed to obtain the stated 4-6 days' aging time.*

Thank you for the comment. The oxidation potential was misrepresented using the $O_3$ concentrations instead of OH. We have modified this discussion as shown below.

L199-L204: "The OH exposure in the OFR was calibrated using the change in carbon monoxide (CO) concentrations inside the OFR, at the same temperature and RH conditions as during the experiments. The change in CO versus light intensity of the lamps uses the CO + OH rate coefficient ($k_{OH+CO}$ ~1.48 * $10^{-13}$ molecules cm$^{-3}$). The residence time in the OFR was ~2.2 min and the average OH exposure in the experiments was ~6.31 * $10^{11}$ molecules sec/cm$^3$ which translated to ~4–6 days of aging under typical atmospheric conditions (OH = 1.5 *$10^6$ molecules cm$^{-3}$)."

*L164-L205 – "It is clear [...] cannot be discounted." - this content seems more appropriate to put in Results & Discussion and/or supplement than in Methods.*

Thank you for the comment. We have significantly modified this section. The paragraphs on the particle size distributions have been moved to the results section 3.1.

L481-L595: "Data from the SMPS and APS downstream of the OFR were merged into a complete size distribution, as shown in Figure 4 for all experiments. The solid lines indicate aerosols generated from real seawater and the dashed lines are those generated from artificial seawater. For sizes < 200 nm the aSSA+SMA distributions were dominated by secondary particle formation, as indicated by their similarity to the SMA-only experiment. This is consistent with previous work that showed only a small fraction of the submicron aSSA+SMA number distribution generated from a MART originated from the pSSA (Mayer et al., 2020b; Prather et al., 2013). Mayer et al., (2020b) also suggested that new particle nucleation was favored over condensation in the OFR due to the high OH concentrations and fast oxidation rates. The SW size distributions had higher concentrations of aSSA+SMA than the ASW at sizes < 100 nm, which may indicate additional gas phase emissions from the seawater, capable of oxidizing to condensable species. In general, both aSSA+SMA distributions agree with the pSSA distributions at larger sizes. However, the convergence of the distributions occurs at different aerosol diameters for the ASW and SW, ~200 nm and 1 μm for the ASW and SW, respectively. In addition, the higher concentration of aSSA+SMA compared to pSSA from natural SW between 200 nm and 1 μm (Figure 4b) suggests some modification of the pSSA. This could occur through a number of different factors, including gas-phase condensation, changes to the seawater microbial activity altering emissions, or minor changes to particle generation due to surface tension or temperature variations.

[revised manuscript text omitted]

by using non-plastic materials. We will nevertheless assume the role of SMA can be interpreted from the ice nucleation experiments, as will be discussed in the subsequent results sections."

We have elected to leave the second paragraph regarding the aerosol phase state where it was. However, this paragraph has been modified as well, as follows.

L242-L401: "The phase state of the particles before entering the CFDC remains uncertain and could affect the heterogeneous nucleating ability of the aerosol population. Several previous studies have analyzed the temperature- and humidity-dependent phase states of NaCl particles, since they are often used as a proxy for SSA. Previous studies have found that the phase state of NaCl after efflorescence depends on the temperature at which efflorescence occurred, with anhydrous NaCl observed for efflorescence temperatures above 273 K (Bartels-Rausch et al., 2021; Koop et al., 2000b; Peckhaus et al., 2016; Wagner et al., 2012). Since the majority of aerosol drying, including the efflorescence of the salt components of pSSA and aSSA+SMA, occurred at room temperature (Figure 2), it is believed the salts would be in the anhydrous form. However, the phase state and morphology of co-emitted organic particles or coatings on the pSSA and aSSA+SMA are less well understood at low temperatures, and the exact structure was uncertain in this study. This is especially difficult to know for the aSSA+SMA particles, due to the exposure to oxidation in the OFR. It is assumed that particles smaller than 200 nm generated from SW contain high fractions of insoluble organic particles or salt particles mixed with organic carbon below 500 nm, similar to previous lab-generated SSA (Bertram et al., 2018; Kaluarachchi et al., 2022a; Prather et al., 2013). In addition, DeMott et al. (2023) showed particle morphologies of laboratory-generated SSA that include organic coatings in the submicron size range, both before and after similar use of an OFR for oxidation studies. The MART was demonstrated to produce substantially similar size distributions and SSA compositions compared to a more natural wave breaking process for bubble bursting (Prather et al., 2013; Stokes et al., 2013), so no bias in organic content in comparison to natural SSA production is expected. During cooling, the RHw would be low enough such that organic particles or coatings may form a glassy state (Ignatius et al., 2016; Knopf et al., 2018; Koop et al., 2011), as shown by the shaded green region in Figure 2, which represents the glass transition conditions for sucrose (Zobrist et al., 2008)."

Aerosols that contain organic material may either enhance heterogeneous nucleation (Wilson et al., 2012) or have no effect (Kasparoglu et al., 2022). For example, organic coatings on mineral dust particles have been shown to suppress heterogeneous nucleation toward higher RH at cirrus temperatures (Möhler et al., 2008), depending also on the coating thickness or the fractional coverage of the particles, while another study found organic coatings on mineral dust had no effect on immersion freezing between 253–233 K regardless of the coating thickness (Kanji et al., 2019). At warmer temperatures > 233 K, organic coatings may be in a less viscous or liquid-like phase state, while at colder temperature may become semi-solid or glassy (Charnawskas et al., 2017), which may explain the differences in ice nucleation behavior of particles with organic coatings in those studies. Specifically, the inclusion of glassy coatings on the pSSA or aSSA+SMA particles may inhibit water uptake and delay dissolution of the salt components until higher RH$_w$ is reached."

[Figure]

**Figure 2.** Expected trajectory and phase state of the pSSA, aSSA+SMA and SMA particles for CFDC experiments, modified from Patnaude et al. (2021). Orange dashed line is the expected efflorescence line for NaCl on the basis of the parametrization of anhydrous NaCl and extrapolated to cirrus temperatures (Tang & Munkelwitz, 1993). The blue shaded region represents the range of possible deliquescence RHs for NaCl and SSA, using the parameterization from Tang & Munkelwitz, (1993) for NaCl extrapolated to colder temperatures as the upper bound and shifting it down 4 % RH for SSA similar to Wagner et al. (2018). The long dashed black lines follow the path of aerosol particles as through drying, cooling, and CFDC scans at different temperatures. The blue circles represent aqueous solutions, gray hexagons represent effloresced pSSA aerosols, and the light blue circles with embedded hexagons represent fully deliquesced particles. The gray hexagons with green outlines and green circles represent the aSSA+SMA and SMA particles, respectively. Lines indicating ice saturation and predicted homogeneous freezing conditions are also denoted. The dotted region represents conditions where aerosol particles experience ice supersaturated conditions and relative humidities that exceed their deliquescence point. The green shaded region represents conditions below the glass transition curve of sucrose from Zobrist et al. (2008).

*L171 – I strongly disagree that these factors alone enhance nucleation. While the gas-phase oxidation rate was indeed accelerated with the use of elevated OH concentrations, gas-to-particle condensation rates could in principle also have been increased by using SSA concentrations and increasing the condensation sink. Since pSSA concentrations were only 140 cm⁻³, (L306), which is far lower than even seed particle concentrations used in many environmental chamber studies with OH concentrations that are closer to atmospheric OH concentrations, it seems to be me that homogenous nucleation of SMA particles was just as likely (if not more likely) due to the very low pSSA condensation sink than the high oxidant concentrations.*

We agree with this analysis. Please see response to the first comment about condensation vs. new particle formation.

*L181 - what exactly constitutes a "blank" or "background" experiment in this context? MART air sampled through the OFR with in the absence of pSSA generation? This was not clear to me.*

The blank experiments were intended to resemble the SMA experiments, where the OFR was turned on but the MART plunging was turned off. Therefore, it was only the MART headspace air that was being sampled. The main difference was for the blank experiment, the MART was filled with DI water. We have modified the sentence to make this clearer.

L578-L580: "For the blank (background) tests, in which the MART was filled with only DI water and the OFR was turned on but MART plunging turned off (black line), it was found that there was also secondary particle formation, likely a result of VOCs emitted from the acrylic material of the MART walls."

*L265-L270: "In these experiments [...] the cycle was repeated" – this content seems more appropriate to put in Methods. It would also be useful to explain why 10% and 0.1% frozen fractions were chosen as the benchmark conditions for ending and beginning the RH scans in the CFDC.*

We moved those sentences to section 2.3 per this recommendation and added a few sentences as to why the ice fraction thresholds were chosen and why we believe they are appropriate for this study.

L421-L429: "The thresholds of ice fraction for the $RH_w$ scans required 10 continuous seconds above 10 % or below 0.1 % in order to reset the scans. Note these activated fractions were calculated based on particle counts in channels above a selected OPC channel to define ice counts during the experiments, as opposed to the more accurate method for calculation of ice fraction that accounted for the background aerosol distribution (described below). Using 10 % and 0.1 % as endpoints of the $RH_w$ scans resulted in a broad range in RHs being covered. By the time 10 % of the particles had nucleated, the $RH_w$ was at above homogeneous freezing conditions (Koop et al., 2000a) and it was not necessary to continue raising the $RH_w$. The lower threshold of 0.1 % for 10 continuous seconds was sufficient to reset the scans as the $RH_w$ was then well below the deliquescence $RH_w$ for NaCl and SSA (~74 %; Tang and Munkelwitz, (1993)), with the exception of $RH_w$ at warmer temperatures > 225 K, where the $RH_w$ scans did not reach below ~75 % but we did not expect heterogeneous freezing conditions."

*L271/Figure 4: Figure 4a is confusing - I don't understand why heterogenous freezing occurs over the first 3 CFDC scans and homogenous freezing occurs over the last 3 CFDC scans for what I assume is nominally the same pSSA. Also, the "Temperature" axis/label is too vague; if I understand Figure 4 correctly, only T_ow is being shown, whereas T_rw is not. Please update the figure axis/legend to clarify this either way.*

The temperature is the column temperature. The reason it rises slightly is due to the $T_{OW}$ increasing. We have modified this figure to make it easier to see the differences in the particles freezing and the RH conditions where this occurred, and added some more descriptive sentences to make this clearer. As explained in the modified text below, a gradual increase in ice particle counts is indicative of heterogeneous freezing, whereas homogeneous freezing is associated with a more abrupt change at a specific $RH_w$.

L597-L604: "Figure 5a shows a time series of CFDC scans for a pSSA experiment. The first three scans, which occurred below 220 K showed a gradual increase in ice particle counts (light blue markers) with increasing $RH_w$, starting around 65 %. The final three scans beginning ~218 K, showed a more modest increase in ice particle counts until the $RH_w$ reached nearly 100 %, resulting in a very sharp increase in ice particle counts. The first three scans indicate particles freezing via heterogeneous nucleation due to the slow and gradual increase of ice particles, with the initial formation of ice particles below 75 % $RH_w$. In

contrast, the last three scans showed very few ice particles below 85 % $RH_w$, but then had a sudden and dramatic spike in ice particles when the $RH_w$ was > 90 %. The fourth scan may be at/near the transition between the two freezing mechanisms, as there were some increased ice particle counts at $RH_w$ < 80 %, but we also observed the sharp spike in ice particles when the $RH_w$ was near water saturation."

[Figure]

**Figure 5.** Time series of CFDC scans for (a) pSSA and (b) SMA experiments generated from real seawater beginning at 204 K. The CFDC column temperature is represented by the red lines, and the blue lines represents the RHw. The black and light blue markers indicate the numbers of total particles counted in the

OPC and those that are considered ice crystals, respectively. The black dotted horizontal lines in the top panels of (a) and (b) represent water saturation.

*L278: What exactly is the "expected homogeneous freezing threshold?"*

Threshold was probably a poor choice of wording. Instead, we have changed this sentence to make this clearer.

L609-L611: "These cases illustrate that SMA froze through a homogeneous freezing mechanism, as there was almost no ice formed until the RH$_w$ exceeded conditions expected for homogeneous freezing (Koop et al., 2000a)."

*L283/Figure 5: why are 1% and 5% frozen fractions shown here, whereas the 10% frozen fraction was used as the RH scan endpoint earlier (Fig. 4)? The presentation of multiple frozen fraction values without explanation for the underlying reasons comes across as confusing/arbitrary.*

See our earlier response as to why the 0.1 and 10 % activation fractions were used. These activation fractions (0.1 % and 10 %) were simply used as automated scan endpoint thresholds to provide a range of RHs for each temperature. Our more accurate method for calculation of ice fraction was only used to determine the 1 % and 5 % ice fraction for presentation. When using the more accurate method described in section 2.3, we did not find observations of 10 % ice fraction. We have added the following to explain.

L620-L621: "Note when using this method, 10 % ice fraction was no longer observed in any of the experiments and is the reason for presentation of 5 % instead."